# Polymyxin B lethality requires energy-dependent outer membrane disruption

Carolina Borrelli [1,2,3,4], Edward J. A. Douglas [1,2], Sophia M. A. Riley[1,2], Aikaterini Ellas Lemonidi[1,2], Gerald Larrouy-Maumus [1,5], Wen-Jung Lu [6], Boyan B. Bonev [6], Andrew M. Edwards [1,2] ✉ & Bart W. Hoogenboom [3,4] ✉

Polymyxin antibiotics target lipopolysaccharides (LPSs) in both membranes of the bacterial cell envelope, leading to bacterial killing through a poorly defined mechanism. Here we demonstrate that metabolic activity is essential for the lethality of clinically relevant doses of polymyxin B (PmB) and leverage this insight to determine its mode of action. PmB killed exponential-phase *Escherichia coli* but did not eliminate stationary-phase cells unless a carbon source was available. Antibiotic lethality correlated with surface protrusions visible by atomic force microscopy and LPS loss from the outer membrane via processes that required LPS synthesis and transport but that were blocked by the MCR-1 polymyxin resistance determinant. While energy-dependent outer-membrane disruption was not directly lethal, it facilitated PmB access to the inner membrane, which the antibiotic permeabilized in an energy-independent manner, leading to cell death. This work reveals how metabolic inactivity confers tolerance of an important, membrane-targeting antibiotic.

The bactericidal activity of many antibiotics is dependent upon cellular metabolic activity[1–5]. For the efficacy of membrane-targeting antibiotics or peptides, however, it has been proposed that metabolic activity is not required, because intact membranes are presumed essential in all physiological states[6–8]. In support of this, many membrane-targeting compounds are effective against metabolically active and inactive bacteria[9–13].

Polymyxin antibiotics are the only membrane-targeting drugs used clinically against Gram-negative bacteria, but often lack efficacy in vivo, despite potent bactericidal activity in vitro[14–18]. Two polymyxins are used therapeutically, polymyxin B (PmB) and polymyxin E (colistin), both of which consist of a cationic cyclic peptide component coupled to an extracyclic peptide chain which in turn is linked to an acyl tail[14,15,18]. Both polymyxin antibiotics target lipopolysaccharides (LPSs) via high-affinity, long-lasting interactions, leading to disruption of the

outer membrane (OM)[18–20]. However, the nature of this OM disruption and the mechanism by which bacteria are killed by the antibiotic is largely unknown[14,15,18,21–24].

Several previous studies have shown that inner-membrane (IM) permeabilization, which is proposed to be required for killing, occurs after OM disruption[14,15,21,25–28]. Our previous work showed that LPS is the IM target of polymyxins, just as it is for the OM[27,28]. While it is unknown how polymyxins cross the OM to access the periplasmic space, the predominant model is 'self-promoted' uptake, by which the cationic properties of the antibacterial disrupt cation bridges between LPS molecules, compromising barrier function and enabling antibiotic ingress through the OM[14,29,30].

Despite the membrane-damaging effect of polymyxins, they do not eradicate persister cells of *Escherichia coli* or *Acinetobacter*

[1]Centre for Bacterial Resistance Biology, Imperial College London, London, UK. [2]Department of Infectious Disease, Imperial College London, London, UK. [3]London Centre for Nanotechnology, University College London, London, UK. [4]Department of Physics & Astronomy, University College London, London, UK. [5]Department of Life Sciences, Imperial College London, London, UK. [6]School of Life Sciences, Queen's Medical Centre, University of Nottingham, Nottingham, UK. ✉e-mail: a.edwards@imperial.ac.uk; b.hoogenboom@ucl.ac.uk

*baumannii*[31,32]. Stationary-phase *E. coli* has also been shown to have a high level of polymyxin tolerance, suggesting that the bactericidal activity of these membrane-targeting antimicrobials is affected by metabolic activity and/or growth state[10]. Accordingly, metabolic activity was found to be required for colistin lethality at clinically relevant doses[13]. However, at concentrations above those typically achieved clinically (>10–50 μg ml$^{-1}$), polymyxin lethality was found to be unaffected by growth phase or metabolic activity[13,32,33].

In summary, there are substantial gaps in our understanding of how polymyxin antibiotics kill bacteria, and the requirement for metabolic activity remains unclear. Therefore, we aimed to determine whether and why metabolic activity was needed for polymyxin lethality and thereby gain a significantly improved understanding of the mechanism by which this class of antibiotics kill bacteria.

## Results

### Metabolic activity affects PmB-mediated membrane disruption and bacterial killing

To assess the role of growth and metabolic activity in PmB-mediated killing, we took equal numbers of *E. coli* in exponential phase (high metabolic activity) or stationary phase (low metabolic activity) and exposed them to a clinically relevant concentration of the antibiotic (4 μg ml$^{-1}$) in minimal medium only (MM) or with glucose (MM + G) to stimulate metabolic activity (Fig. 1a and Extended Data Fig. 1a)[34–36]. Exponential-phase *E. coli* cells were rapidly killed by PmB regardless of whether glucose was present (Fig. 1a), and PmB efficiently killed stationary-phase bacteria in the presence of glucose, albeit after a lag phase of 15 min during which there was no decrease in viability (Fig. 1a). Strikingly, however, there was no detectable killing of stationary-phase *E. coli* during exposure to PmB in the absence of glucose, indicating that metabolic activity is required for the bactericidal activity of the antibiotic (Fig. 1a). This effect of glucose was dose dependent, and the non-metabolizable glucose analogue 2-deoxyglucose did not support PmB-mediated killing of stationary-phase *E. coli* (Extended Data Fig. 1b,c). Conversely, there does not appear to be a requirement for cell division for PmB killing, as MM without antibiotic (±glucose) did not support noticeable growth of *E. coli* from either exponential or stationary phase during the assay.

Using atomic force microscopy (AFM) to examine the effect of the antibiotic on the OM of living bacteria[37,38], we tracked the effects of the antibiotics in real time: PmB exposure of exponential-phase *E. coli* resulted in rapid and profound changes to the cell surface, with the appearance of numerous protrusions, regardless of the presence of glucose (Fig. 1b). By contrast, stationary-phase *E. coli* only showed PmB-induced OM changes in the presence of glucose (Fig. 1b), correlating with the effects on bacterial viability (Fig. 1a,b). Scanning electron microscopy replicated the AFM observations, confirming these distinct, glucose-dependent phenotypes of stationary-phase cells exposed to PmB, while the antibiotic caused surface protrusions in exponential-phase *E. coli* regardless of the presence of glucose (Extended Data Fig. 2).

Next, using the small hydrophobic dye NPN (*N*-phenyl-1-naphthylamine)[39], we showed that the binding of PmB to its lipid A target was unaffected by glucose or growth phase (Fig. 1c and Supplementary Fig. 1a). This finding was supported by mass spectrometry[27], which ruled out the presence of lipid A modifications in stationary-phase cells that might reduce PmB binding but intriguingly did show a large reduction in LPS abundance in cells exposed to PmB in MM + G (Extended Data Fig. 3). We further investigated changes to OM integrity by measuring the leakage of heterologously expressed mCherry (28 kDa) from the periplasm[40]. In the presence of PmB and glucose, stationary-phase *E. coli* showed a significant increase of mCherry in the supernatant, compared with cells not exposed to the antibiotic from 90 min onwards. This suggests that minor OM permeabilization/disruption occurs rapidly, as measured by NPN, but that major

disruption to the OM—as measured by mCherry release—occurs much more slowly, possibly via lysis (Fig. 1d).

Finally, we used SYTOX green nucleic acid stain, to assess permeability of both the OM and IM during PmB exposure. As expected from the viability data (Fig. 1a), stationary-phase *E. coli* showed significantly increased permeability in the presence of glucose, after a lag of 15 min (Fig. 1e); permeability also occurred in the absence of glucose, but at significantly lower levels than for cells exposed to the antibiotic in the presence of glucose (Fig. 1e). Again, consistent with the viability data (Fig. 1a), PmB-induced IM permeability was unaffected by glucose in exponential-phase *E. coli* (Supplementary Fig. 1b).

Taken together, these data show that PmB binds to *E. coli* regardless of metabolic state, leading to minor OM disruption sufficient to allow NPN ingress, but that bactericidal activity, OM protrusions and release of periplasmic protein require metabolic activity. It is worth noting that stationary-phase viability and OM/IM permeation were similarly dependent on glucose across a diverse panel of laboratory and clinical isolates, indicative of a broadly conserved phenotype (Extended Data Fig. 4).

### PmB-mediated OM damage results in significant LPS loss

High-resolution AFM imaging of the OM of stationary-phase *E. coli* showed a similar abundance and organization of OM protein networks as previously reported[37,38] (Fig. 2a and Extended Data Fig. 5; note porous appearance of the OM in phase images). The porin networks were initially unaffected by PmB, but over time, the appearance of the PmB-exposed OM was dominated by protrusions at a scale >10 nm (Fig. 2a and Supplementary Fig. 5b,c). On these living *E. coli* cells, we did not observe any PmB-induced LPS rearrangement into hexagonal lattices, such as previously reported based on AFM on solid-supported, collapsed OM vesicles[22–24].

Further examination of PmB-induced OM changes revealed that LPS was released from cells exposed to PmB in the presence of glucose but not in the absence of the antibiotic or glucose (Fig. 2b). Analysis of bacterial cells found a corresponding loss of LPS from *E. coli* exposed to PmB in the presence but not absence of glucose (Fig. 2c,d and Supplementary Fig. 2), which was maximal by 15 min and fully consistent with the mass spectrometry results reported above (Extended Data Figs. 3b and 6a). This coincided with the appearance of surface protrusions, which subsequently increased in abundance, up to IM permeabilization as measured by SYTOX (Supplementary Fig. 3). Moreover, LPS loss at 15 min was not due to cell lysis as, in contrast to LPS, there were no differences in levels of the cytoplasmic protein GroEL or OM porin OmpF across our experimental conditions (Fig. 2c).

Combined, these data revealed that PmB at least initially leaves the OM protein networks intact but causes substantial and rapid LPS loss in the presence of glucose. PmB-induced LPS loss also occurred with *Pseudomonas aeruginosa* and with exponential-phase *E. coli* in Mueller–Hinton broth (MHB), indicative of a conserved process (Extended Data Fig. 6 and Supplementary Fig. 4).

### PmB-mediated LPS loss and bacterial killing require energy and LPS synthesis

The observed dependence on metabolic activity for PmB lethality suggests an energy-dependent bacterial response to PmB challenge. As PmB-triggered LPS release required metabolic activity, we hypothesized that LPS synthesis and/or transport to the OM contributed to bacterial killing.

To establish energy dependence, we sought to distinguish between carbon- and energy-related effects. Using equimolar concentrations of various carbon sources in MM, we measured stationary-phase *E. coli* survival under PmB challenge and, in parallel experiments without PmB, bacterial ATP production and overall LPS abundance (Fig. 3a). In these experiments, bacterial PmB susceptibility correlated with

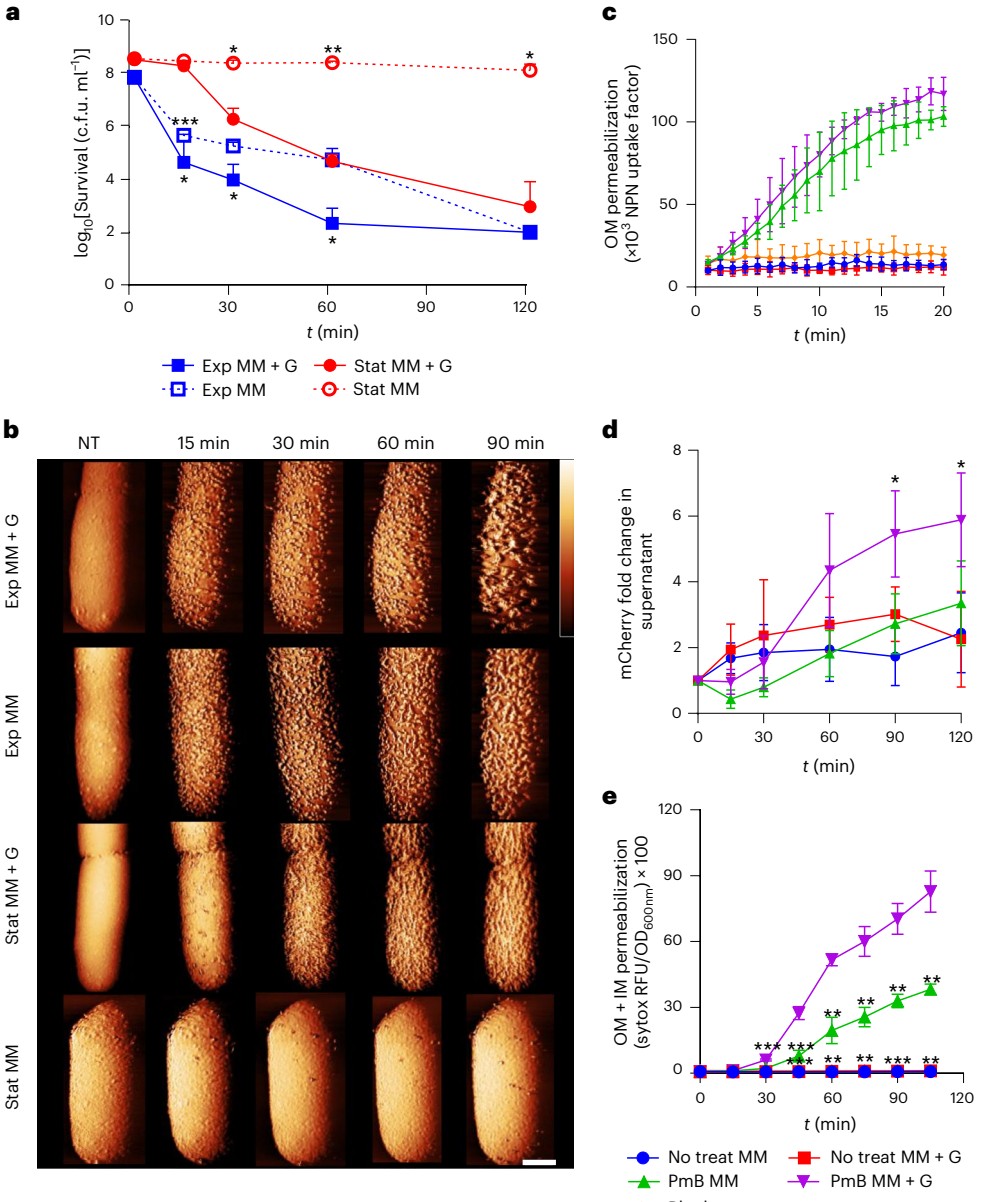

**Fig. 1 | PmB lethality requires metabolic activity and is associated with significant morphological changes to the cell surface. a**, Survival of exponential-phase (Exp) or stationary-phase (Stat) *E. coli* exposed to 4 µg ml⁻¹ PmB in MM ± glucose, as determined by c.f.u. counts. **b**, AFM phase images showing exponential- or stationary-phase *E. coli* cells exposed to 2.5 µg ml⁻¹ PmB in MM ± glucose, shown as a function of time. Scale bar, 250 nm. Colour phase scale (scale insert in first row of image at *t* = 90 min), 6 degrees (row 1), 4 degrees (row 2), 5 degrees (rows 3 and 4). Except for stationary-phase cells without glucose, all bacteria were SYTOX positive by the end of the imaging (Supplementary Fig. 6a). It is worth noting that a lower PmB concentration was used for AFM relative to other experiments, due to the low density of cells used in these assays. The 'phase' in degrees represents the shift in the phase of oscillation of the AFM cantilever. **c**, OM disruption of stationary-phase *E. coli* cells during the first 20 min of exposure to 4 µg ml⁻¹ PmB, as determined by uptake

of NPN fluorescent dye. **d**, OM disruption of *E. coli* _Peri_mCherry as determined by egress of the fluorescent protein mCherry into the culture supernatants of stationary-phase bacteria exposed or not to 4 µg ml⁻¹ PmB in MM ± glucose. **e**, Combined OM and IM disruption of stationary-phase *E. coli* exposed to 4 µg ml⁻¹ PmB in MM ± glucose, as determined by uptake of the fluorescent nucleic acid dye SYTOX green. RFU, relative fluorescence units. For **c** and **e**, the blank value refers to the relevant fluorophore in medium without bacteria. For **c** and **e**, 'No treat MM' refers to the absence of PmB in minimal medium; 'No treat MM + G' refers to the absence of PmB in minimal medium containing 0.36% glucose. All experiments were replicated in *n* = 3 independent assays. Error bars show the standard deviation of the mean. Significant differences were determined between stationary-phase MM + G with PmB and each of the other conditions by two-way repeated measures ANOVA. *$P < 0.05$; **$P < 0.01$; ***$P < 0.001$; ****$P < 0.0001$.

ATP production and with the amount of LPS associated with cells in the absence of PmB (Fig. 3a,b and Extended Data Fig. 7). In the presence of PmB, the amount of LPS dropped to lower levels in the presence of carbon sources that yielded higher ATP production (Fig. 3a,b and Extended Data Fig. 7), confirming the importance of energy for PmB-induced LPS loss and bacterial killing.

To establish a link between LPS production and PmB-induced LPS loss and killing, we performed PmB killing assays in the presence of various inhibitors[41]. Inhibition of UDP-3-O-acyl-N-acetylglucosamine deacetylase (LpxC), the rate-limiting step of lipid A synthesis, reduced the rate and degree of PmB-mediated killing of both *E. coli* and *P. aeruginosa*, whereas the protein synthesis inhibitor tetracycline had no effect

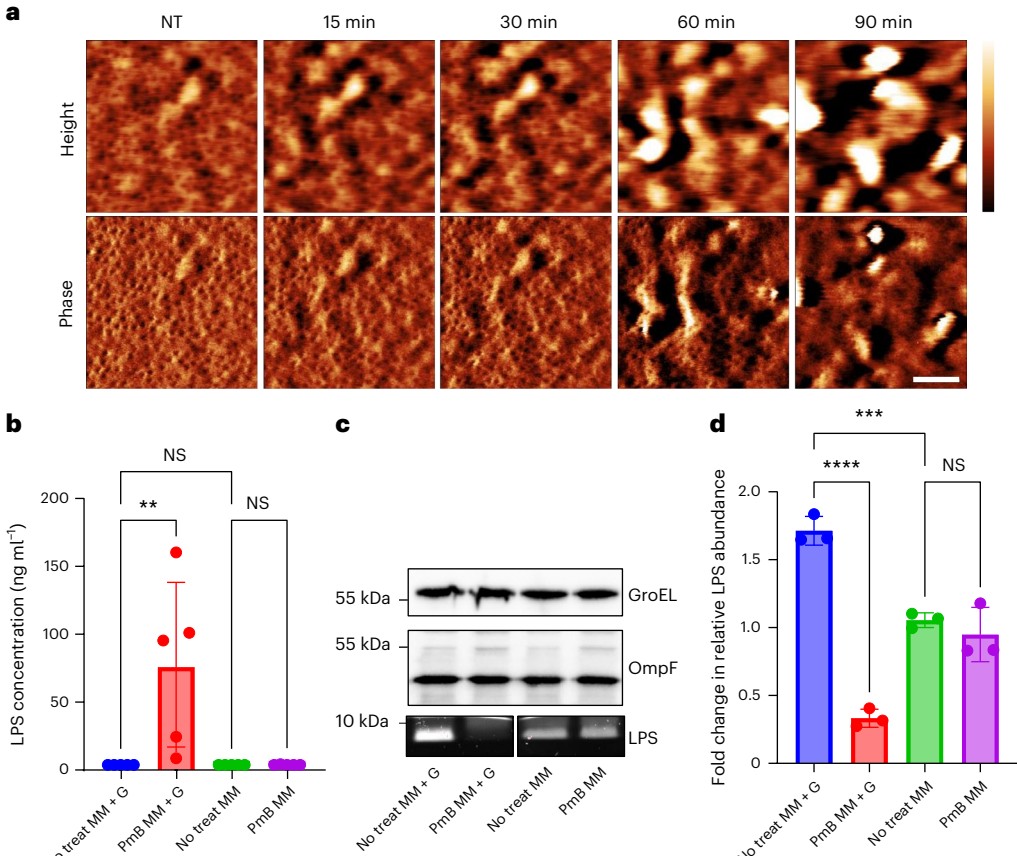

**Fig. 2 | PmB-mediated OM disruption results in LPS loss without detectable disruption of the porin network. a**, High-magnification AFM height and phase images of stationary-phase *E. coli* exposed to 2.5 µg ml⁻¹ PmB in MM + G, showing the porin network remains largely intact for at least 60 min of antibiotic challenge. Scale bar, 50 nm. Height scale (scale inset in first row of image at *t* = 90 min), 3 nm. Phase scale, 1 deg. **b**, Kdo analysis of LPS recovered from filtered supernatants of stationary-phase *E. coli* exposed, or not, to 4 µg ml⁻¹ PmB in MM ± glucose for 15 min (*n* = 5). **c**, Levels of OmpF, GroEL and total LPS in *E. coli*

exposed, or not, to 4 µg ml⁻¹ PmB in MM ± glucose for 15 min. **d**, Quantification of LPS levels in *E. coli* exposed, or not, to 4 µg ml⁻¹ PmB in MM ± glucose for 15 min. For **b**–**d** 'No treat MM' refers to the untreated condition absence of PmB in minimal medium; 'No treat MM + G' refers to the untreated condition absence of PmB in minimal medium containing 0.36% glucose. Unless stated otherwise, experiments were replicated in *n* = 3 independent assays. Error bars show the standard deviation of the mean. Statistical significance of differences was determined by one-way ANOVA; ***$P < 0.001$; ****$P < 0.0001$; NS, not significant.

(Fig. 3c–e and Extended Data Fig. 8a–c). There was also a slower progression of PmB-triggered surface protrusions in the presence of CHIR-090 (an LpxC inhibitor) relative to PmB alone (Extended Data Fig. 8d). We also found that an inhibitor of the LPS transporter MsbA (multicopy suppressor of the HtrB temperature-sensitive phenotype) reduced killing by PmB and slowed protrusion appearance, while a strain with reduced LPS transport protein D (LptD) function was also killed more slowly than the wild type (Fig. 3f,g and Extended Data Fig. 8e). For all inhibitors, PmB binding to the OM was not affected, as determined via NPN uptake, whereas reduction of LPS synthesis and transport significantly reduced PmB-induced IM permeation as measured by SYTOX ingress into the cytoplasm (Supplementary Figs. 5 and 6).

Further evidence for the importance of LPS transport came from experiments with novobiocin, which promotes LPS trafficking from the IM to the OM[42] and was found to promote PmB-mediated LPS loss (Extended Data Fig. 9). This finding also provides a molecular explanation for the potentiating effect of novobiocin on polymyxin susceptibility[42,43].

Combined, these experiments revealed that the magnitude of PmB-mediated LPS loss correlated with metabolic activity, which in turn correlated with bacterial killing. The role of metabolic activity in LPS loss was due, at least in part, to a requirement for LPS synthesis and transport.

## LPS loss from the OM enables PmB-mediated IM disruption and bacterial killing

To differentiate between PmB action on the OM and on the IM, we examined whether PmB could kill metabolically active cells by LPS loss alone, without the need for permeabilization of the IM. For this purpose, we first exposed *E. coli* to PmB for 15 min in MM + G, long enough for LPS loss to reach its maximum but before IM disruption occurred and before colony-forming unit (c.f.u.) counts started to decrease relative to the inoculum (Fig. 1a,e and Extended Data Fig. 6a,b). We then washed the bacteria to remove unbound antibiotic, which also removed released LPS, followed by incubation in MM ± PmB and ±glucose. In the absence of PmB, bacteria survived at the same level over 2 h in the presence or absence of glucose (Fig. 4a). By contrast, bacteria that were re-exposed to PmB were efficiently killed, regardless of the presence of glucose (Fig. 4a). Therefore, PmB-mediated LPS loss alone is not sufficient to kill bacteria. Moreover, the requirement for metabolic activity for PmB-mediated killing appeared to only apply to LPS loss from the OM and is not required for subsequent IM disruption and killing.

To further test the impact of LPS loss on bacterial killing, we exposed stationary-phase *E. coli* to the nonlethal PmB nonapeptide (PmBN)[18], which also caused significant LPS loss within 15 min in the presence of glucose, and to EDTA (ethylenediamine tetraacetic acid), which caused LPS loss both in presence and absence of glucose

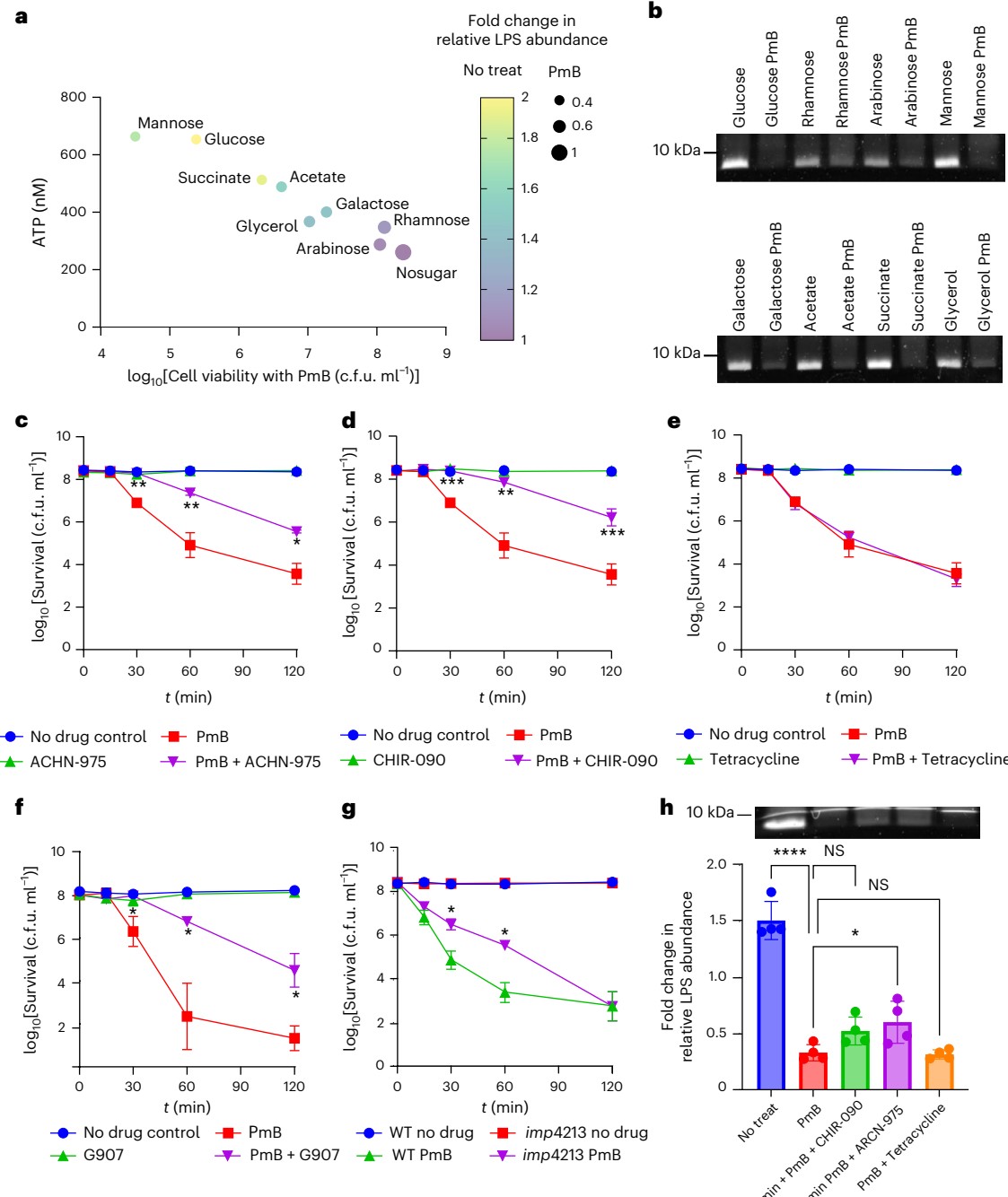

**Fig. 3 | PmB-triggered LPS loss and bacterial killing require ATP and LPS synthesis and transport. a**, Four-way representation of the effect of various carbon sources, or none, on ATP production and LPS levels in stationary-phase *E. coli* in the absence of PmB, and bacterial survival and LPS levels after exposure to 4 µg ml⁻¹ PmB. The more yellow the symbol, the more LPS is produced in the absence of PmB; the smaller the symbol, the more LPS that is lost during PmB exposure. **b**, SDS–PAGE analysis of LPS from stationary-phase *E. coli* exposed to various carbon sources in MM for 15 min in the absence or presence of 4 µg ml⁻¹ PmB. **c–f**, Survival of stationary-phase *E. coli* in MM + G supplemented, or not, with PmB, with or without growth-inhibitory concentrations (1× MIC) LpxC inhibitors ACHN-975 (**c**) or CHIR-090 (**d**) or protein synthesis inhibitor tetracycline (**e**) or MsbA inhibitor G907 (**f**). **g**, Survival of *E. coli* wild type (WT) or

LptD4213 strain exposed, or not, to 4 µg ml⁻¹ PmB. **h**, LPS levels of non-treated *E. coli* or bacteria exposed to 4 µg ml⁻¹ PmB ± LpxC inhibitors (CHIR-090, ARCN-975) or tetracycline for 15 min (*n* = 4). Panel includes an SDS–PAGE image of LPS band intensity. For **h** 'No treat' refers to the untreated condition absence of PmB in minimal medium containing 0.36% glucose. It is worth noting that the experiments with G907 used *E. coli* LptD4213 to enable antibiotic ingress. Unless otherwise stated, all experiments were replicated in *n* = 3 independent assays. Error bars show the standard deviation of the mean. Statistical significance of differences was determined by one-way (**h**) or by two-way repeated measures ANOVA between the PmB-treated and PmB + antibiotic-treated groups (**c–f**) or between wild type with PmB and *imp*4213 with PmB (**g**). *P < 0.05; **P < 0.01; ***P < 0.001; ****P < 0.0001; NS, not significant.

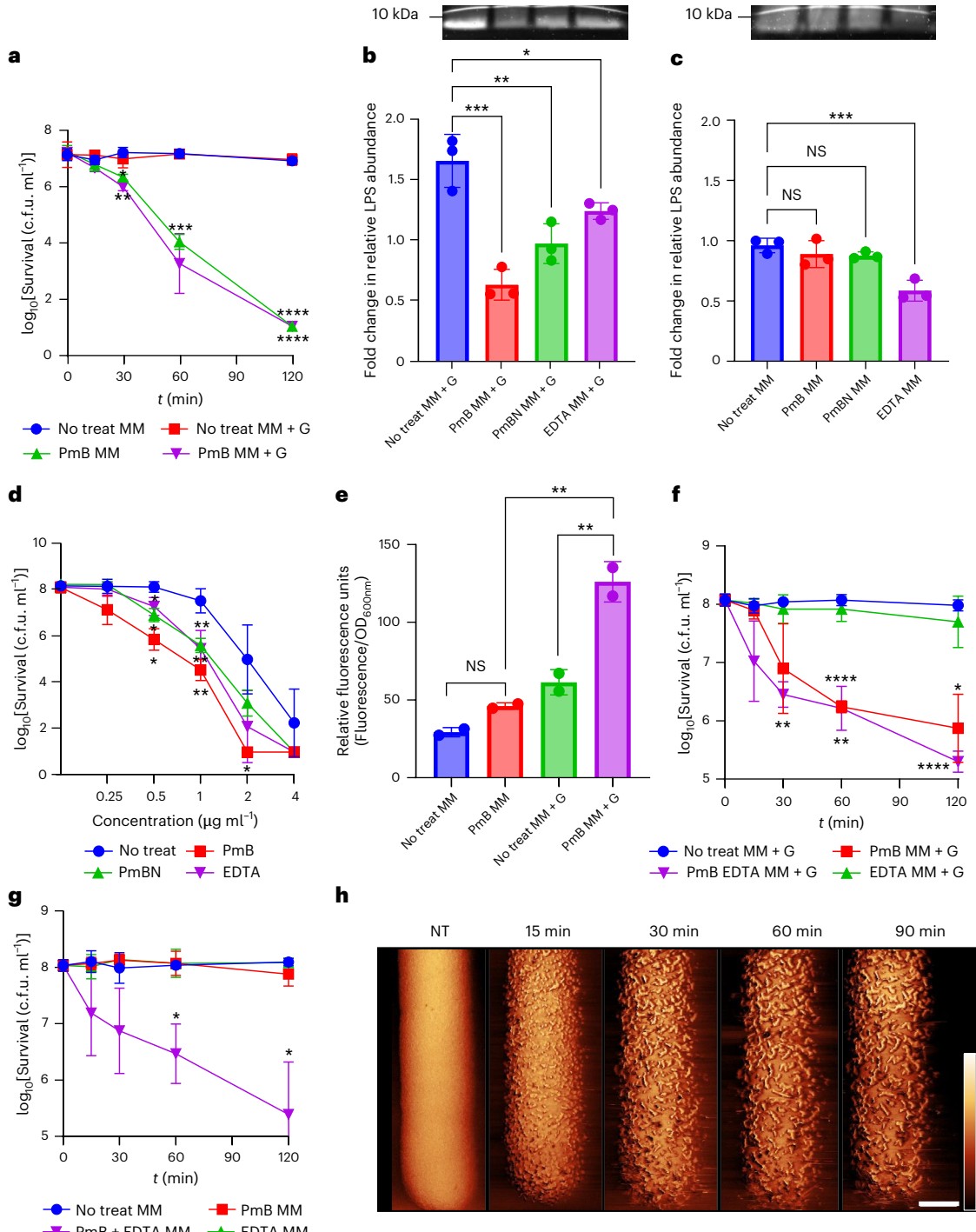

**Fig. 4 | PmB-mediated LPS loss is necessary for lethality because it provides the antibiotic with access to the IM. a**, Survival (c.f.u.) of stationary-phase *E. coli* first pre-treated with 4 µg ml⁻¹ PmB in MM + G for 15 min before washing to remove unbound antibiotic, and next (from *t* = 0) exposed to 4 µg ml⁻¹ PmB, or not, in MM ± glucose. **b**, LPS levels of untreated *E. coli* or bacteria exposed to 4 µg ml⁻¹ PmB, 4 µg ml⁻¹ PmBN or 10 mM EDTA for 15 min in MM + G. Note that experiments with EDTA used MM that was not supplemented with MgSO₄ or CaCl₂. Panel includes SDS–PAGE image of LPS band intensity. **c**, LPS levels of non-treated *E. coli* or bacteria exposed to 4 µg ml⁻¹ PmB, 4 µg ml⁻¹ PmBN or 10 mM EDTA for 15 min in MM. Panel includes SDS–PAGE image of LPS band intensity. **d**, Survival of stationary-phase *E. coli* exposed to a range of PmB concentrations for 2 h in MM + G, following a 15 min pre-treatment, or not, with 4 µg ml⁻¹ PmB, 4 µg ml⁻¹ PmBN or 10 mM EDTA in MM + G. **e**, Levels of bodipy-vancomycin

labelling of *E. coli* exposed, or not, to 4 µg ml⁻¹ PmB in MM ± glucose for 15 min. **f**, Survival of *E. coli* exposed, or not, to PmB with or without 10 mM EDTA in MM + G. **g**, Survival of *E. coli* exposed, or not, to PmB with or without 10 mM EDTA in MM. **h**, AFM phase images showing stationary-phase *E. coli* exposed to 2.5 µg ml⁻¹ PmB and 10 mM EDTA in MM. Scale bar, 250 nm; phase scale (scale inset in image at *t* = 90 min), 5 deg. The same cell was imaged during the experiment. All experiments were replicated in *n* = 3 independent assays. Error bars show the standard deviation of the mean. Statistical significance of differences was determined by one-way (in **b**, **c** and **e**) or two-way repeated measures ANOVA between No treat MM + G and PmB MM + G and between No treat MM and PmB MM (in **a**) between No treat and pre-treatment conditions (in **d**) between No treat and antibiotic conditions (in **f** and **g**). *$P < 0.05$; **$P < 0.01$; ***$P < 0.001$; ****$P < 0.0001$; NS, not significant.

(Fig. 4b,c). At the doses tested, neither EDTA nor PmBN had any effect on bacterial viability and did not trigger membrane protrusions (Extended Data Fig. 10 and Supplementary Fig. 7).

Based on these findings, we hypothesized that polymyxin-triggered LPS release, while not bactericidal on its own, facilitates antibiotic entry into the periplasm, enabling PmB to access and permeate the IM and thereby kill the cell. To test this, we exposed stationary-phase E. coli to PmB, PmBN or EDTA for 15 min in MM + G to trigger LPS loss, as above, before removing these OM-disrupting agents (and released LPS) by washing, and finally measured bacterial survival in the presence of freshly added PmB. Each of the three pre-treatments enabled killing by PmB at concentrations below those required to kill untreated control cells, indicating that LPS loss sensitizes bacteria to PmB-mediated killing (Fig. 4d).

It is worth noting that we saw a greater level of bacterial killing for experiments where bacteria were pretreated with a membrane-disrupting agent before washing and subsequent exposure to PmB (for example, Fig. 4a,d), compared with experiments without a pre-treatment step (for example, Fig. 3c,d,f). A possible explanation for this difference is that released LPS sequesters PmB, reducing its activity[44]. As this is washed away in the pre-treatment experiments, it cannot reduce the activity of the subsequent PmB dose. Furthermore, once the OM is compromised by the membrane-disrupting pre-treatment, subsequent PmB treatment can efficiently access the IM. Combined, this explains the higher levels of bacterial killing in the pre-treatment experiments relative to other assays.

To demonstrate that PmB-triggered LPS loss causes the OM to become permeable to molecules of the size of PmB, we showed periplasmic ingress of a bodipy-conjugated analogue of the glycopeptide antibiotic vancomycin (Mw = 1723.35 g mol$^{-1}$ versus 1301.56 g mol$^{-1}$ for PmB) (Fig. 4e)[45]. However, minor permeabilization of the OM, as occurs with PmB in the absence of glucose, did not permit entry of vancomycin into the periplasm (Fig. 4e).

As EDTA caused LPS loss from stationary-phase E. coli in the absence of glucose (Fig. 4c), we hypothesized that EDTA would enable PmB to access the IM and thereby kill E. coli in the absence of energy. First, in MM + G, there was a significant loss of viability of E. coli exposed to PmB alone and to PmB combined with EDTA, but not EDTA alone (Fig. 4f). However, levels of bacterial viability were the same for cells exposed to PmB alone and PmB–EDTA, indicating this combination was not synergistic at the concentrations used (Fig. 4f). We then repeated this experiment in MM without glucose. In keeping with our earlier findings, metabolically inactive cells were not killed by PmB alone (Fig. 4g). However, there was a significant loss of viability of metabolically inactive cells exposed to PmB and EDTA together (Fig. 4). In addition, analysis of EDTA/PmB co-treatment of metabolically inactive cells by AFM revealed a roughening of the bacterial surface, with numerous protrusions that resembled the effects observed of PmB alone in the presence of glucose (Fig. 4h and Supplementary Fig. 8).

Together, these experiments show that metabolic activity is required for PmB lethality because it supports LPS release, which provides the antibiotic with access to the IM, the disruption of which leads to bacterial killing. However, if LPS loss occurs via a PmB-independent mechanism, such as EDTA, then there is no need for metabolic activity, showing that IM disruption occurs via an energy-independent process.

### Polymyxin resistance determinant MCR-1 blocks PmB-mediated LPS release

Mobile colistin resistance (mcr) genes, which encode phosphoethanolamine (pEtN) transferases, are globally disseminated in the Enterobacterales[46–48]. MCR-1, which is the most common variant, selectively modifies the 4′ phosphate of lipid A targeted by PmB, reducing susceptibility to polymyxin antibiotics[47]. Previous work showed that MCR-1 protected the IM of both whole cells and spheroplasts from colistin[28,39]. However, previous studies have shown that

polymyxin antibiotics can disrupt the OM of polymyxin-resistant E. coli as evidenced by ingress of NPN or hydrophobic antibiotics such as rifampicin[14,28,39]. As such, the impact of MCR-1 on OM integrity during PmB exposure is unclear. Based on the work above, we hypothesized that pEtN modification would contribute to polymyxin resistance by inhibiting PmB-triggered LPS release, which would block access of the antibiotic to the IM.

To test this, we used E. coli transformed with an inducible mcr-1 construct but allowed expression to occur at basal levels (that is, no inducer) so that we had minimal production of MCR-1 and thus only a 4-fold increase in PmB minimal inhibitory concentration (MIC) above the wild-type strain (Supplementary Table 1). Despite this modest change in susceptibility, MCR-1 fully protected E. coli from PmB at 4 μg ml$^{-1}$ in MM + G, whereas there was a >4-log$_{10}$ reduction in c.f.u. counts of the wild type (Fig. 5a). There was also a substantial reduction in the viability of E. coli that produced a catalytically inactive version of MCR-1 (MCR-1*) (Fig. 5a)[49]. As expected, MCR-1 reduced but did not prevent OM disruption as indicated by NPN uptake relative to the wild-type strain, whereas the IM was fully protected from PmB (Fig. 5b,c)[27,28,39,49].

In support of our hypothesis, we next showed that MCR-1 prevented PmB-triggered LPS loss, as well as surface protrusions, both of which occurred in the wild type and the strain expressing the catalytically inactive MCR-1 variant (Fig. 5d–f).

Therefore, MCR-1 mediated pEtN modification of lipid A prevents LPS loss from E. coli exposed to PmB, which contributes to resistance by blocking access of the antibiotic to the IM. However, it is not clear whether the lack of LPS release is due to reduced PmB binding or greater membrane integrity due to the pEtN modification or a combination of both factors.

## Discussion

The mechanism(s) by which polymyxin antibiotics kill bacteria has been the focus of multiple studies, with various conclusions reached and models proposed[14,15,18,21,22,50]. In this work, we established that metabolic activity is required for bactericidal activity of a clinically relevant concentration of PmB and used this as a starting point to build a refined model for the mechanism by which polymyxin antibiotics kill bacteria (Fig. 6). We found that the initial interaction of PmB with the OM was unaffected by the metabolic state of the bacteria, explaining how polymyxins can sensitize metabolically dormant persister bacteria to antibiotics[32,51,52] (Fig. 6a). In metabolically inactive bacteria, there is no further disruption to the bacterial surface and no loss of viability. However, in metabolically active E. coli, PmB causes rapid, large-scale LPS loss, which required ATP-dependent LPS synthesis and transport (Fig. 6b). With the loss of OM integrity, PmB gains access to LPS in the IM, and surface protrusions appear, although it is not yet clear whether these processes are directly linked (Fig. 6c). Finally, the interaction of PmB with LPS in the IM results in IM disruption and bacterial killing via an energy-independent process, and OM disruption increases, sufficient to allow the egress of proteins from the periplasm (Fig. 6d).

The full mechanism of PmB-triggered LPS loss remains to be elucidated. While EDTA and PmB both disrupt the cation bridges that stabilize LPS molecules in the OM, only EDTA causes LPS release in the absence of metabolic activity. This likely reflects the fact that EDTA chelates cations, whereas PmB displaces cations from LPS by binding lipid A phosphates. While cation displacement by PmB weakens inter-LPS interactions, it appears that this is only sufficient to cause minor permeabilization. We speculate that the insertion of PmB into the LPS monolayer causes slipping between molecules, and as more LPS is transported into the OM, this leads to loss of PmB-bound LPS from the OM. Such slipping could be consistent with the complex interplay between PmB and OM fluidity[53,54].

Several antibiotic classes require bacterial metabolic activity for lethality[2–5]. However, our finding that this requirement also applies to PmB goes against the common assumption that the efficacy of

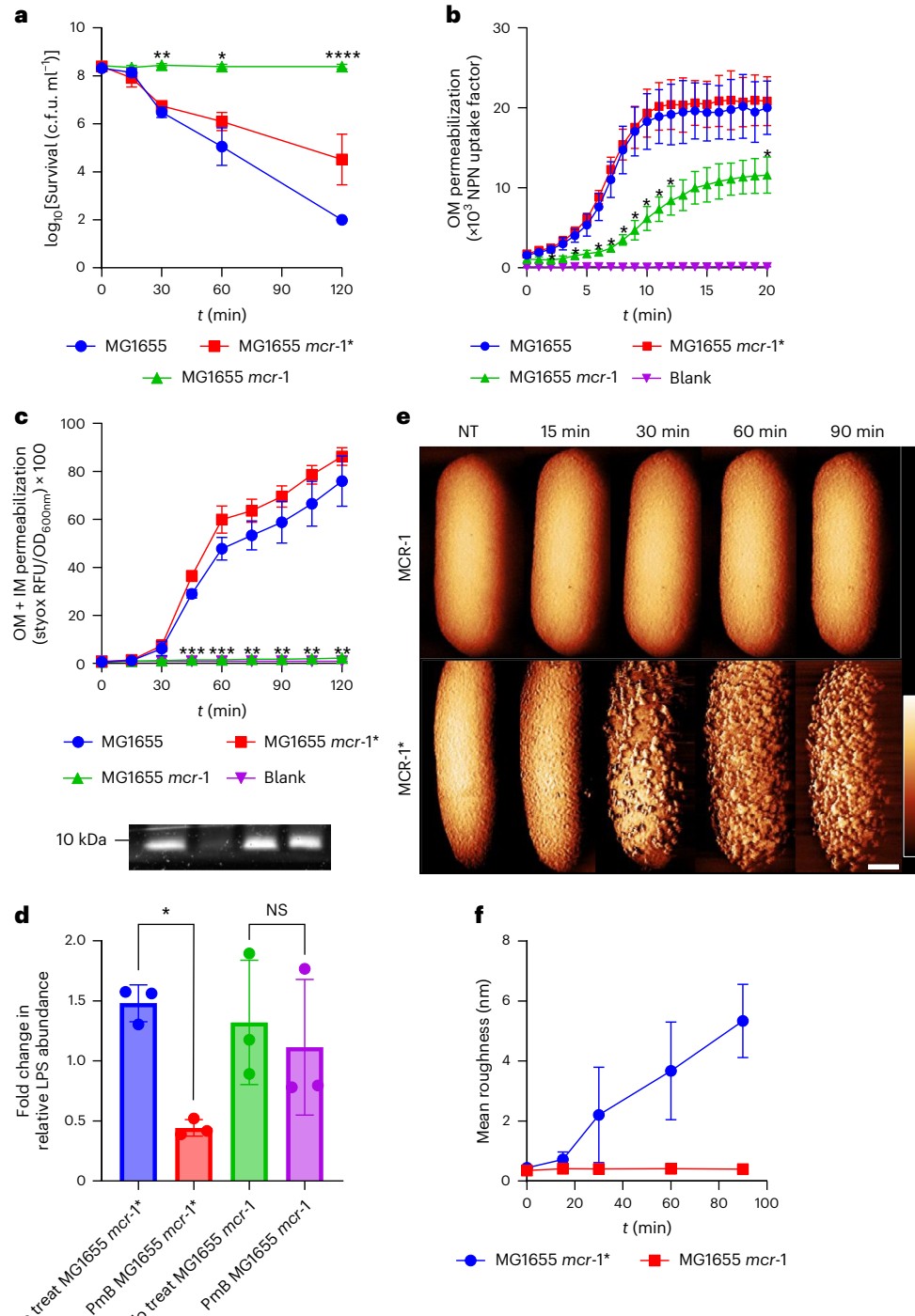

**Fig. 5 | MCR-1 prevents PmB-triggered LPS release. a**, Survival of stationary-phase MG1655, MG1655 *mcr*-1* or MG1655 *mcr*-1 exposed to 4 µg ml⁻¹ PmB in MM + G, as determined by c.f.u. counts. **b**, OM disruption of stationary-phase MG1655, MG1655 *mcr*-1* or MG1655 *mcr*-1 cells during the first 20 min of exposure to 4 µg ml⁻¹ PmB in MM + G, as determined by uptake of NPN fluorescent dye. **c**, OM and IM disruption of stationary-phase MG1655, MG1655 *mcr*-1* or MG1655 *mcr*-1 exposed to 4 µg ml⁻¹ PmB in MM + G, as determined by uptake of the fluorescent dye SYTOX green. **d**, Total LPS in stationary-phase MG1655 *mcr*-1* or MG1655 *mcr*-1 exposed, or not, to 4 µg ml⁻¹ PmB in MM + G for 15 min. Graph shows quantification of LPS levels from three independent experiments.

**e**, AFM phase images showing stationary-phase MG1655 *mcr*-1* and MG1655 *mcr*-1 exposed to 2.5 µg ml⁻¹ PmB in MM + G. The same cell was imaged during the experiment. Scale bar, 250 nm; phase scale (scale inset in second row of image at *t* = 90 min), 4 deg. **f**, Mean surface roughness of MG1655 *mcr*-1* and MG1655 *mcr*-1 treated with 2.5 µg ml⁻¹ PmB in MM + G. For **b** and **c**, the blank value refers to the relevant fluorophore in medium without bacteria. All experiments were replicated in *n* = 3 independent assays. Error bars show the standard deviation of the mean. Statistical significance of differences was determined by one-way (in **d**) or two-way repeated measures ANOVA between MG1655 and *mcr*-1* or *mcr*-1 (in **a**, **b** and **c**). *$P < 0.05$; **$P < 0.01$; ***$P < 0.001$; ****$P < 0.0001$; NS, not significant.

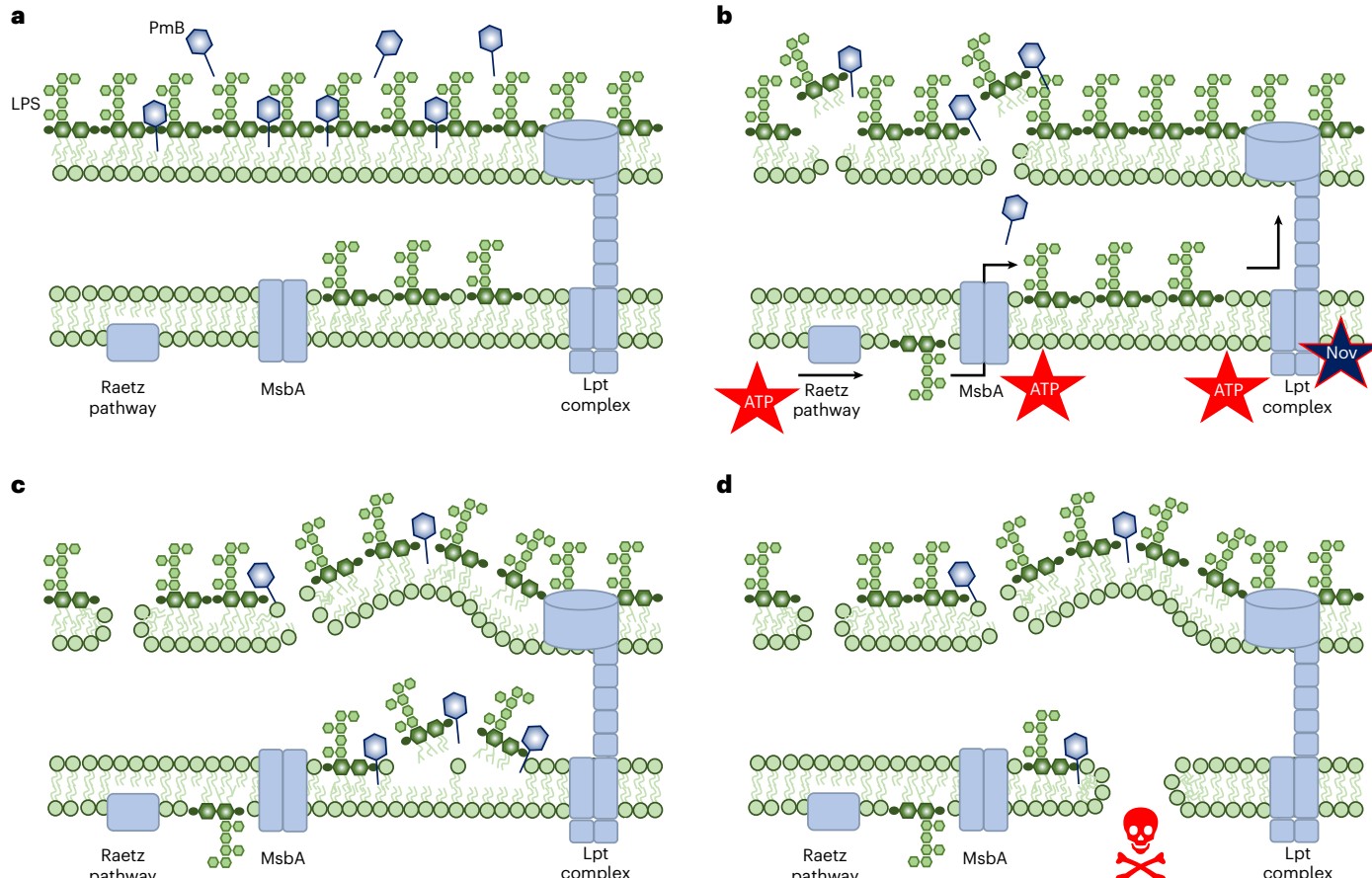

**Fig. 6 | Proposed model for how PmB kills bacteria. a**, PmB binds to the OM, causing minor permeabilization, regardless of bacterial metabolic activity (insufficient damage to provide PmB with access to the periplasm). **b**, In metabolically active bacteria, PmB triggers LPS release via a process that requires ATP-dependent synthesis (Raetz pathway) and transport of LPS (further boosted by novobiocin (Nov)). **c**, The shedding of LPS compromises the integrity of the OM, resulting in surface protrusions, and enables ingress of the antibiotic into the periplasm where it targets LPS in the IM, leading to permeabilization. **d**, The loss of integrity of both the OM and IM results in bacterial killing.

membrane-targeting antibacterials is independent of metabolic activity. It also provides an explanation for polymyxin tolerance observed in stationary-phase cells and persisters[4,5,9,10,31,32,55,56]. An important distinction, though, is that while many bactericidal antibiotics kill via metabolism-dependent production of reactive metabolites[5,8,57], the key energy-dependent step required for polymyxin activity is disruption of the OM, which enables the antibiotic to permeabilize the IM and kill the bacterium.

We have previously shown that sub-inhibitory concentrations of the LptD inhibitor murepavadin promoted polymyxin susceptibility of *P. aeruginosa*, via LPS accumulation in the IM, without reducing LPS transport to the OM[27]. Based on the work presented here, we would predict that growth inhibitory concentrations of inhibitors of the Lpt system would reduce killing by polymyxins. However, we could not test this because the available inhibitors (murepavadin and thanatin) are cationic peptides that cause OM disruption, including membrane blebbing, which would compromise the findings[58–60]. Instead, we capitalized on the observation that novobiocin promoted LPS transport by enhancing ATPase activity, which in turn increased bacterial susceptibility to polymyxin antibiotics, although it was not known why at the time[42,43].

Although our conclusions agree with some previous reports, they may appear inconsistent with others. For example, there is evidence that PmB is active against metabolically inactive bacteria. However, those studies used either exponential-phase cells that may retain a degree of metabolic activity or used assay conditions with low concentrations of nutrients or supra-physiological concentrations of polymyxins where specificity for LPS is lost due to the amphipathic nature of these antibiotics[13,15,32–35,61]. There have also been conflicting studies on the contribution of reactive oxygen species to bactericidal activity of PmB[62–65]. Our finding that energy is required for severe OM disruption does not rule out a role for reactive oxygen species, which could arise via LPS synthesis, for example. Finally, while we show surface protrusions, it is unclear whether these are released as vesicles, which has been reported to occur at sub-inhibitory PmB concentrations[20,66–71]. More work is also needed to determine whether PmB-induced vesicles are driven exclusively by antibiotic action on the membrane or part of bacterial stress response. For example, previous work indicated that a trigger of the Sigma E (RpoE) membrane stress response system resulted in increased vesicle production[72], and there is evidence that the conjugative plasmid expression (Cpx) and reactive chlorine species (Rcs) systems, which respond to IM and OM stress, respectively, are activated by polymyxin exposure and contribute to bacterial survival[73–77]. As such, there remains much to understand about how bacteria respond to polymyxin antibiotics and how this influences survival.

Despite the loss of LPS that occurred during PmB exposure, cells remained viable until IM disruption occurred, in agreement with our and others' previous work showing that IM damage is required for lethality[25,27,52]. Although the OM is an important load-bearing structure, much of the mechanical strength is provided by the porin network,

which forms tight associations with LPS[78–81] and which remained largely intact during PmB exposure (Fig. 3a). The mechanical integrity of the porin network may therefore explain how *E. coli* can tolerate major LPS loss without a reduction in viability[38,78].

Intrinsic and some acquired polymyxin resistance is associated with very high levels of lipid A modification with cationic pEtN and/or 4-amino-4-deoxy-l-arabinose (L-ara4N) moieties, which prevents PmB binding to the OM[14,18,20,48]. However, MCR-mediated resistance often involves modification of only some of the lipid A in the OM, which likely explains how PmB can bind and sufficiently permeabilize the OM for the ingress of hydrophobic antibiotics such as rifampicin[28,39]. We found that some OM permeabilization occurred for *E. coli mcr*-1 but also showed that the MCR-1-mediated lipid A modification prevented PmB-triggered LPS release, which in turn prevented access of the antibiotic to the IM. This, coupled with the protection of the IM via LPS modification[27,28], provides distinct layers of protection against PmB.

Minimal media, including M9 used in these studies, have been found to better predict antibiotic susceptibility of bacteria at infection sites than rich laboratory media, which likely reflects the reduced metabolic activity of organisms in vivo[1,2,82,83]. That said, as M9 does not fully replicate the host environment, further work will be needed to comprehensively understand the relationship between host environment, bacterial metabolism and polymyxin treatment outcomes. For example, one aspect of infection absent in these studies is host immunity. Previous work has shown that polymyxins synergize with complement and the antibacterial histone H2A[84,85]. PmB has been shown to reduce the pro-inflammatory activity of LPS, which likely modulates the host immune response[86].

In summary, this work advances our understanding of the mechanism of action of polymyxin antibiotics by showing that the nature and magnitude of PmB-mediated OM damage is dependent upon the metabolic activity of the cell, which in turns modulates access of the antibiotic to the IM, disruption of which is required for lethality but is not energy dependent (Fig. 6). This work also provides an explanation for how polymyxin tolerance occurs in persister and stationary-phase cells, despite the membrane-disrupting activity of these antibiotics (Fig. 6).

## Methods

### Bacterial strains and growth conditions

The bacterial strains used in this study are listed in Supplementary Table 1 (refs. [87–94]). *E. coli* strains expressing *mcr*-1 or *mcr*-1* under the control of the arabinose-inducible *araBAD* (pBAD) promoter were based on those described previously and were transformed with plasmids synthesized by Thermo Fisher[48]. Unless stated otherwise, all strains were grown in MHB (Millipore) for 18 h at 37 °C with shaking (180 r.p.m.) to stationary phase. Where indicated, exponential-phase bacteria were generated by inoculating fresh MHB with 1:1,000 dilution of stationary-phase cells followed by a second growth period of 3 h at 37 °C with shaking (180 r.p.m.) For all experimentation, bacteria were washed three times with 1× M9 minimal medium (Gibco) supplemented with 2 mM MgSO$_4$ and 0.1 mM CaCl$_2$[37] (referred to here as minimal medium, MM). Washed cells were then resuspended in MM ± 0.36% glucose (MM + G) to 10$^8$ c.f.u. ml$^{-1}$. To enumerate bacterial c.f.u., 10-fold serial dilutions were made in PBS and plated onto Mueller–Hinton agar (MHB supplemented with 1.5% bacteriological agar). Inoculated agar plates were incubated statically for 18 h in air at 37 °C.

### Determination of antibiotic MICs

The MIC of PmB, the LpxC inhibitors CHIR-090 and ACHN-975, tetracycline, the MsbA inhibitor G907, novobiocin and murepavadin against indicated bacterial strains was determined according to the well-established broth microdilution method[95]. In brief, a 96-well microtitre plate was used to prepare a range of antibiotic concentrations in 100 µl of MM + G or cation-adjusted MHB via a series of 2-fold serial dilutions. For antibiotic synergy determination, checkerboard analyses were performed by preparing 2-fold serial dilutions of two antibiotics, with each one across a different axis, generating an 8 × 8 matrix to assess the MICs of each antibiotic when used in combination[96]. Stationary-phase bacteria were diluted to 1 × 10$^6$ c.f.u. ml$^{-1}$ in MM + G or cation-adjusted MHB and 100 µl used to seed each well of the microtitre plate to give a final inoculation density of 5 × 10$^5$ c.f.u. ml$^{-1}$. Plates were then incubated statically at 37 °C for 18 h in air, at which point the MIC was defined as the lowest antibiotic concentration at which there was no visible growth of bacteria. In some cases, the extent of bacterial growth after incubation was also determined by obtaining measurements of optical density at 595 nm (OD$_{595nm}$) using a Bio-Rad iMark microplate absorbance reader (Bio-Rad Laboratories).

### Antibiotic time-kill assays

Following three washes in MM, stationary-phase or exponential-phase bacteria were added at a final concentration of 1 × 10$^8$ c.f.u. ml$^{-1}$ to 3 ml of MM, MM + G or MM without cations and supplemented with 10 mM EDTA, containing PmB (4 µg ml$^{-1}$) when required. This assay was also repeated in the presence of PmB and 1× MIC of a range of different antibiotics (CHIR-090, ACHN-975, tetracycline, G907), as well as in MM supplemented with a selection of equimolar concentrations of either glucose, rhamnose, arabinose, mannose, galactose, acetate, succinate or glycerol. Cultures were incubated at 37 °C with shaking (180 r.p.m.). At each time point (0, 15, 30, 60 and 120 min), aliquots were taken, serially diluted 10-fold in PBS and plated to determine bacterial viability by determination of c.f.u. counts.

### AFM sample preparation

Bacterial imaging by AFM followed protocols described previously[37,38]. Briefly, for sample preparation, round glass coverslips of 13 mm diameter were used. They were washed by sonication in 1% *w/v* SDS for 10 min at 37 kHz and 100% power, then rinsed with MilliQ water and ethanol and dried with a nitrogen gun. Then they were plasma cleaned for 2 min at 70% power. These two procedures were repeated once more. Then coverslips were coated with Vectabond: they were placed in a beaker with a 1:50 ratio of Vectabond solution to acetone (40 ml of acetone and 800 µl Vectabond solution) for 5 min, then rinsed with MilliQ water and dried with a nitrogen gun[97]. The coverslips were glued to microscope slides with water-soluble glue purchased from Bruker (Reprorubber thin pour, Flexbar). Subsequently, *E. coli* was cultured to stationary phase (18 h) in MHB, then diluted 1:100 into fresh MHB and incubated to mid-exponential phase. Bacteria were then prepared by centrifuging 3 times at 17,000 × *g* for 90 s and resuspended in MM or MM + G at an OD$_{600}$ of 0.5. (MM: 1× M9 salts (ThermoFisher Scientific), 2 mM MgSO$_4$, 0.1 mM CaCl$_2$; MM + G: MM with 0.36% glucose). They were resuspended the fourth time in 100 µl 20 mM HEPES and incubated onto the Vectabond-coated glass coverslips for 5 min. They were then washed off with imaging media (MM or MM + G) to remove unadhered bacteria. SYTOX green nucleic acid stain (at a final concentration of 5 µM) was added and incubated at room temperature for 5 min. We first imaged untreated bacteria in MM or MM + G, then with a pipette this was removed as much as possible and replaced with either PmB, EDTA, PmBN, CHIR-090 or combinations, added in the same way. Scans were subsequently taken after the introduction of the drug.

### AFM imaging and analysis

All experiments were performed in dynamic mode with a Nanowizard III AFM with UltraSpeed head (Bruker AXS), with an Andor Zyla 5.5 USB3 fluorescence camera on an Olympus IX 73 inverted optical microscope. FastScanD cantilevers were used, with a resonant frequency around 110 kHz and a spring constant of 0.25 N m$^{-1}$. The drive frequency used ranged from 95 to 120 kHz, depending on the cantilever resonance measured by calibration. The setpoint was between 10 and 15 nm (approximately 50% to 70% relative to free amplitude). The whole-cell

images were acquired at 2 to 2.5 Hz line frequencies, $2 \times 2 \ \mu m^2$, and $512 \times 512$ or $512 \times 256$ pixels. Higher-magnification images were acquired over $500 \times 500 \ nm^2$ and $512 \times 512$ pixels, recorded at 4 to 5 Hz line frequencies. Images in Fig. 2a were acquired over $200 \times 200 \ nm^2$ and $512 \times 512$ pixels recorded at 8 to 10 Hz line frequency. To process the images obtained from AFM the software Gwyddion 2.65 was used (https://gwyddion.net/)[98].

The following steps were taken to obtain a post-processed image: (1) level data by mean plane subtraction; (2) level by flattening base; (3) if there were strongly protruding features, apply a mask by selecting features above a 50% threshold, to prevent these from biasing the background correction and row alignment; (3) align rows by second polynomial line-by-line background subtraction (and exclude the masked region if applicable); (4) apply a 1–2 pixel Gaussian filter to the whole image to remove high-frequency noise (remove mask if applicable); and (5) define zero reference and adjust the colour scale to highlight key features.

### Scanning electron microscopy

The *E. coli* cells were cultured in MHB to stationary phase before bacteria were collected using centrifugation, and the cell pellet was washed three times with M9 medium without glucose. The cells were then diluted to $10^8$ c.f.u. ml$^{-1}$ in M9 ± glucose with 4 µg ml$^{-1}$ PmB, and samples were collected at various time points through centrifugation for further analysis. The samples obtained were treated with 2.5% glutaraldehyde in a 0.1 M sodium cacodylate buffer (pH = 7.3) and were allowed to fix overnight at 4 °C. After fixation, the cells were suspended in the 0.1 M sodium cacodylate buffer and subjected to dehydration using ethanol concentrations of 10%, 20%, 30%, 40%, 50%, 70%, 80% and 100%, with a 10 min incubation at each step. The drying procedure was performed via critical point drying, and a platinum coating of 10 nm thickness was applied after the samples were completely dried. The prepared samples were examined under a scanning electron microscope (Zeiss Crossbeam 550 FIB-SEM) at an accelerating voltage of 2 kV, and all images were analysed after conversion for differences in magnification. The ImageJ/Fiji software (version 1.53) was used in scanning electron microscope image processing and analysis.

### OM permeability assay by NPN uptake

To detect OM damage by PmB, the well-established NPN uptake assay was performed[39]. The fluorescent probe NPN (Acros Organics), at a final concentration of 10 µM, was added to 100 µl of $2 \times 10^8$ c.f.u. ml$^{-1}$ stationary-phase or exponential-phase bacteria in MM, MM + G or MM without cations and supplemented with 10 mM EDTA in a black microtitre plate with clear-bottomed wells (Greiner Bio-One). Relevant concentrations of PmB or PmB in combination with 1× MIC of various antibiotics were added to the wells of the microtitre plate to give a total volume of 200 µl. Fluorescence, at 37 °C with shaking, was measured immediately using a Tecan Infinite 200 pro plate reader using an excitation wavelength of 355 nm and an emission wavelength of 405 nm. Fluorescence measurements were obtained every min for 20 min, and the degree of OM permeabilization, referred to as the NPN uptake factor, was calculated using the following formula:

$$\frac{\text{Fluorescence of sample with NPN} - \text{Fluorescence of sample without NPN}}{\text{Fluorescence of MM with NPN} - \text{Fluorescence of MM without NPN}}$$

The resulting NPN uptake factor values were normalized according to OD$_{600nm}$.

### OM permeability assay by mCherry egress

To further assess OM damage by PmB, $_{Peri}$mCherry *E. coli* was prepared by transforming the pPerimCh plasmid[40] into *E. coli* MG1655. This plasmid encodes a constitutively expressed mCherry construct that accumulates in the periplasm. Stationary-phase bacteria ($2 \times 10^8$ c.f.u. ml$^{-1}$)

were incubated with PmB in MM or MM + G. At each time point, 200 µl of the culture was centrifuged to separate bacteria and culture medium, and the supernatant was removed and placed in the wells of a black clear-bottomed 96-well plate (Greiner Bio). The fluorescence was measured with an excitation wavelength of 587 nm and an emission wavelength of 610 nm.

### IM permeability assay

To measure IM permeabilization by PmB, the well-established SYTOX green assay was used[99]. Following washes in MM, stationary-phase bacteria at an inoculum density of $1 \times 10^8$ c.f.u. ml$^{-1}$ was added to 3 ml of MM or MM + G containing the relevant antibiotics. The fluorescent probe SYTOX green (Invitrogen) was added to these cultures at a final concentration of 1 µM. Aliquots (200 µl) were transferred to a black microtitre plate with clear-bottomed wells (Greiner Bio-One). Fluorescence and OD$_{600nm}$ were measured in a Tecan Infinite 200 Pro plate reader (excitation at 535 nm, emission at 617 nm) every 15 min for 2 h at 37 °C with shaking. Raw fluorescence readings were normalized according to OD$_{600nm}$.

### Determination of lipid A modification by mass spectrometry

Lipid A was prepared and analysed for modifications by mass spectroscopy as described previously[27,28]. Briefly, lipid A was detached from carbohydrates via mild acid hydrolysis (2% acetic acid, 100 °C 30 min), washed in $H_2O$, loaded on the target and overlaid with 9H-Pyridol[3,4-B] indole (Norharmane, Sigma-Aldrich) used at 10 mg ml$^{-1}$ in chloroform to methanol 9:1 and mixed before air drying and matrix-assisted laser desorption ionization time-of-flight analysis using a MALDI Biotyper Sirius system (Bruker Daltonics).

### Detection of LPS by sodium dodecyl sulfate–polyacrylamide gel electrophoresis

The experimental conditions for LPS quantification were the same as those used for the antibiotic time-kill assay. Briefly, $1 \times 10^8$ c.f.u. ml$^{-1}$ of stationary-phase bacteria was added to 3 ml cultures of MM or MM + G in the presence of either PmB, PmB with 1× MIC of various antibiotics, PmBN or EDTA. LPS quantification was also performed in the presence of PmB in MM supplemented with a selection of equimolar concentrations of either glucose, rhamnose, arabinose, mannose, galactose, acetate, succinate or glycerol. Following 15 min incubation at 37 °C with shaking (180 r.p.m.), samples were pelleted at $16,000 \times g$ and washed twice with PBS. Pellets were incubated with 2% SDS and DNAse1 and incubated for 10 min at room temperature. The pellet was resuspended in 100 µl of 1× sample buffer (4% β-mercaptoethanol, 10% glycerol, 0.1 M Tris, 2% SDS at pH 6.8). The volume of sample buffer was adjusted according to the fold change in OD$_{600nm}$ compared to time point zero to give equal biomass between samples. Next, samples were incubated for 10 min at 95 °C. They were then checked to contain equal biomass by separation on a 12% Tris–glycine gel, followed by Coomassie staining. The samples were then incubated with 100 µg ml$^{-1}$ proteinase K overnight at 55 °C to remove protein and again separated on a 12% Tris–glycine gel. The gel was fixed overnight in a solution of 50% methanol and 5% acetic acid. LPS was then quantified using the Pro-Q Emerald 300 Lipopolysaccharide Gel Stain Kit (Invitrogen) according to the manufacturer's protocol. Densitometric analysis was performed using Fiji ImageJ with the resulting values used to interpolate the standard curve performed in Supplementary Fig. 8c, to calculate relative LPS abundance. Undigested total protein samples served as loading controls.

### Western blotting

GroEL and OmpF were detected by western blotting of total bacterial protein samples, obtained as described above for detection of LPS by sodium dodecyl sulfate–polyacrylamide gel electrophoresis (SDS–PAGE), using the following primary antibodies: Monoclonal anti-GroEL antibodies were obtained from Abcam (clone, EPR28718-8; catalogue

number AB318970, lot number 1096886-3) and used at 1:2,000 dilution; polyclonal Anti-OmpF antibodies were obtained from Invitrogen (catalogue number PAS-121442, lot number 2K4557550) and used at 1:5,000 dilution. Bound primary antibodies were detected using polyclonal Goat Anti-Rabbit HRP conjugated (catalogue number AB6721, lot number GR342221), used at a 1:2,000 dilution, and ECL western blotting detection reagent (Amersham, RPN, 2106).

## Detection and quantification of released LPS

This protocol was provided by R. Hancock (https://cmdr.ubc.ca/bobh/method/kdo-assay/). Briefly, 50 µl of supernatant (generated from 15 min incubation of $1 \times 10^8$ c.f.u. ml$^{-1}$ of stationary-phase bacteria in MM ± glucose, and ±4 µg ml$^{-1}$) was boiled in 50 µl 0.5 N H$_2$SO$_4$ for 15 min, followed by the sequential addition of 200 µl arsenite reagent (0.3 µM NaAsO$_2$ dissolved in 0.5 N HCl) and 800 µl 42 mM 2-thiobarbituric acid. The solution was boiled for a further 10 min before the addition of 1.5 ml butanol reagent (5.0 ml of concentrated HCl added to 95 ml $n$-butanol). The butanol layer was separated from the solution by centrifugation for 5 min at $2,500 \times g$. The Kdo (3-deoxy-D-manno-octulosonic acid) extinction coefficient (OD$_{552nm}$ − OD$_{509nm}$) was calculated by measuring 200 µl for absorbance at OD$_{552nm}$ and OD$_{509nm}$ in a Tecan Infinite 200 Pro plate reader. The difference in OD was used to interpolate a standard curve generated by performing Kdo analysis on a 1/10 serial dilution of 5 mg ml$^{-1}$ of purified rough LPS (Supplementary Fig. 8a).

## ATP quantification

To quantify ATP levels of stationary-phase bacteria in different carbon sources conditions, an ATP standard curve was first generated using the BacTiter-Glo (Promega) kit. A 10-fold serial dilution series of ATP was performed in MM over a concentration range of 1,000 to 0.01 nM in a white microtitre plate (Greiner Bio-One). The BacTiter-Glo buffer was added to the substrate to make the BacTiter-Glo substrate. This substrate was combined with the dilution series, and luminescence was measured in a Tecan Infinite 200 Pro plate reader using a 100 ms integration time. Following this, stationary-phase bacteria at an inoculum density of $1 \times 10^8$ c.f.u. ml$^{-1}$ was added to MM or MM containing a range of equimolar concentrations of different carbon sources. At each time point (0, 15 and 30 min), 100 µl of the culture was combined with 100 µl of the substrate in a white microtitre plate, and luminescence was measured. ATP concentrations under these different conditions were interpolated from the standard curve.

## Measuring vancomycin binding

Following three washes in MM, stationary-phase bacteria were added at a final concentration of $1 \times 10^8$ c.f.u. ml$^{-1}$ to 3 ml of MM ± glucose and ±4 µg ml$^{-1}$ PmB, in the presence of 10 µg ml$^{-1}$ BODIPY FL Vancomycin (Invitrogen). Cultures were incubated at 37 °C for 15 min, followed by three washes in PBS to remove unbound BODIPY FL Vancomycin. Cultures were resuspended in 3 ml PBS and 200 µl added to a black clear-bottomed 96-well plate (Greiner Bio). Fluorescence and OD$_{600nm}$ were measured in a Tecan Infinite 200 Pro plate reader (excitation at 480 nm, emission at 520 nm). Raw fluorescence readings were normalized according to OD$_{600nm}$.

## Statistics and reproducibility

Experiments were performed on at least three independent occasions, and the resulting data are presented as the arithmetic mean of these biological repeats unless stated otherwise. Error bars, where shown, represent the standard deviation of the mean unless stated otherwise. For single comparisons, a two-tailed Student's $t$-test was used to analyse the data. For multiple comparisons at a single time point or concentration, data were analysed using one-way analysis of variance (ANOVA) or Kruskal–Wallis test. Where data were obtained at several different time points or concentrations, two-way ANOVA was used for statistical analyses. Appropriate post hoc tests (Dunnett's, Tukey's, Sidak's,

Dunn's) were carried out to correct for multiple comparisons, with details provided in the figure legends. Asterisks on graphs indicate significant differences between data. Precise $P$ values for all comparisons are provided in the source data files. All statistical analyses were performed using GraphPad Prism 7 software (GraphPad Software).

Where images are shown for example gels, blots or AFM, these are representative examples of at least three replicates, unless otherwise stated.

## Reporting summary

Further information on research design is available in the Nature Portfolio Reporting Summary linked to this article.

## Data availability

All data supporting the conclusions of this work are available via the UCL Research Data Repository at https://doi.org/10.5522/04/29282072.v1. Source data are provided with this paper.

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

## Acknowledgements

S. Rutherford and K. Buchholz (Genentech), T. Clarke and S. Stoneham (Imperial College London), and G. Benn (University College London/University of Oxford) are thanked for helpful discussions. This work was funded by Biotechnology and Biological Sciences Research Council (award BB/Y003667/1 to A.M.E. and G.L.-M.; BB/X002446/1, BB/X001547/1 and BB/X000370/1 to B.B.B., B.W.H. and A.M.E., respectively) and by the Wellcome Trust (227923/Z/23/Z to B.W.H.). C.B. is supported by the Centre for Doctoral Training in the Advanced Characterisation of Materials, funded by the Engineering and Physical Sciences Research Council and Science Foundation Ireland (EP/S023259/1). We acknowledge Biotechnology and Biological Sciences Research Council (BB/R000042/1 to B.W.H.) and Engineering and Physical Sciences Research Council (EP/K031953/1, via the Interdisciplinary Research Centre in Early-Warning Sensing Systems for Infectious Diseases) for funding equipment. A.M.E. is supported in part by the National Institute for Health and Care Research Imperial Biomedical Research Centre.

## Author contributions

C.B., E.J.A.D, W.-J.L., B.B.B., A.M.E. and B.W.H. designed the research; C.B., E.J.A.D, S.M.A.R., A.E.L., W.-J.L., G.L.-M. and A.M.E. performed the research; C.B., E.J.A.D, S.M.A.R., A.E.L., W.-J.L., B.B.B., A.M.E. and B.W.H. analysed data; C.B., E.J.A.D, B.B.B., A.M.E. and B.W.H. wrote the paper. C.B. and E.J.A.D contributed equally. A.M.E. and B.W.H contributed equally.

## Competing interests

B.W.H. holds an executive position at AFM manufacturer Nanosurf. Nanosurf did not play any role in the design or execution of this study. The other authors declare no competing interests.

## Additional information

**Extended data** is available for this paper at https://doi.org/10.1038/s41564-025-02133-1.

**Correspondence and requests for materials** should be addressed to Andrew M. Edwards or Bart W. Hoogenboom.

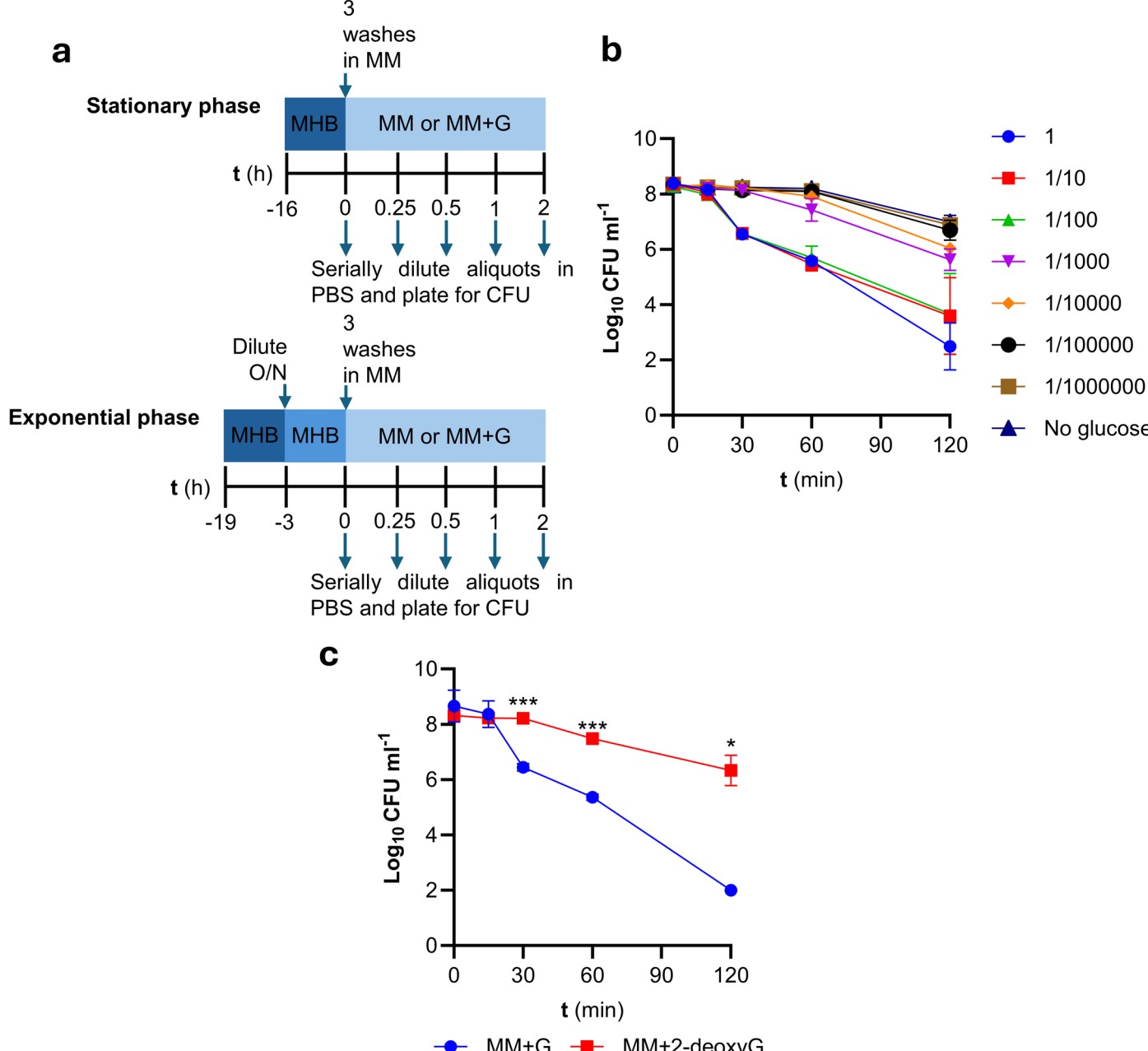

**Extended Data Fig. 1 | Polymyxin B lethality requires metabolic activity.**
**a**, Visual representation of the set-up for all microbiological assays. *E. coli* MG1655 was grown in 3 ml of MHB for 16 h at 37 °C 180 r.p.m. to stationary phase. The culture was washed three times in MM and resuspended at an inoculum density of 1×10⁸ in 3 ml of MM or MM + G. To test exponential phase *E. coli*, the overnight culture was diluted 1/1000 in fresh MHB and grown for a further 3 h. **b**, Survival of stationary phase *E. coli* exposed to 4 µg ml⁻¹ PmB from t = 0 in the presence of a 1/10 dilution series of 0.36% glucose. A 1/100 dilution of 0.36% glucose supported PmB killing, however, a 1/1000 dilution substantially reduced the rate and degree of PmB killing. **c**, Survival of stationary phase *E. coli* exposed to 4 µg ml⁻¹ PmB from t = 0 in MM + G or MM with an equimolar concentration of 2-deoxy-D-glucose, as determined by CFU counts. All experiments were replicated in n = 3 independent assays. Error bars show the standard deviation of the mean. Significant differences were determined by two-way repeated measures ANOVA. P=*<0.05, ***<0.001.

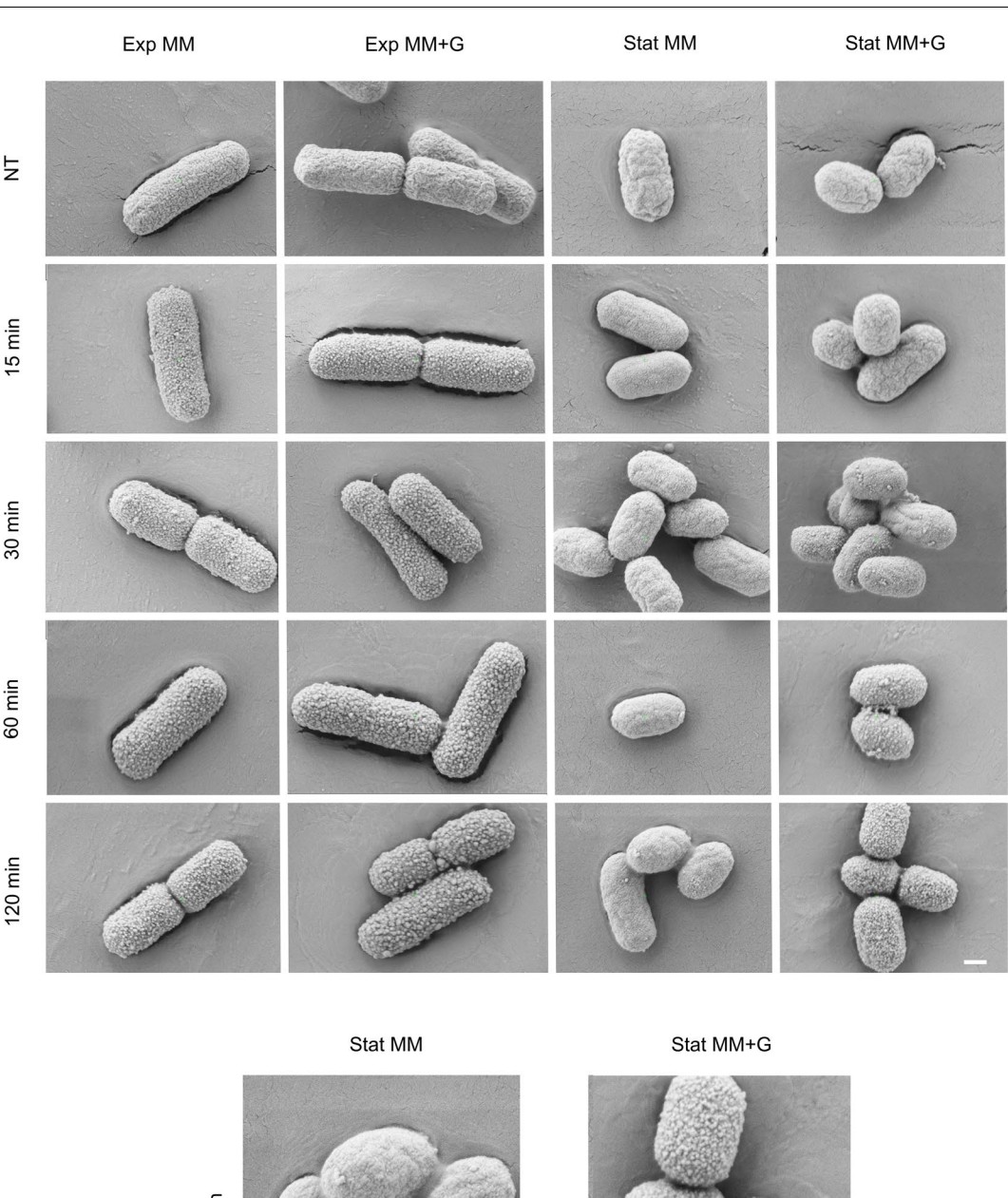

**Extended Data Fig. 2 | PmB causes surface protrusions on exponential phase *E. coli* MG1655 and only when glucose is present in stationary phase cells.** Exponential (Exp) or – (Stat) *E. coli* cells were exposed to PmB (4 µg ml⁻¹) in MM ± glucose and samples taken at the indicated time points. In keeping with the data from AFM studies, stationary phase cells exposed to PmB in the presence of glucose began to show surface protrusions from ~30 min, which increased over time. Conversely, stationary phase cells exposed to the polymyxin without glucose showed minimal to no surface protrusions. PmB triggered surface protrusions in exponential phase cells whether glucose was present or not. Images for stationary phase cells at 120 min are shown again in an enlarged image to highlight differences in surface appearance. Scale bar represents 400 nm (N = 1).

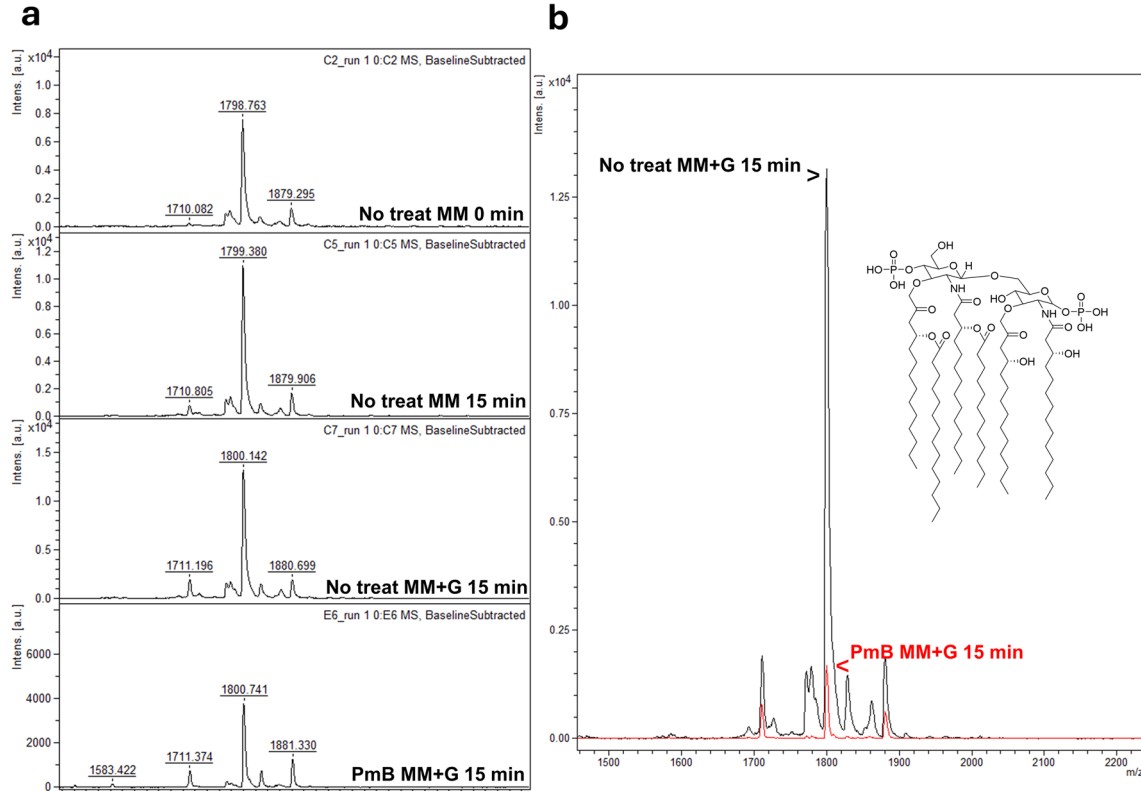

**Extended Data Fig. 3 | Tolerance to PmB is not associated with modifications to lipid A. a**, From top to bottom, representative mass spectra showing unmodified lipid A from stationary phase *E. coli* at time 0 min in MM, stationary phase *E. coli* at time 15 min in MM, and stationary phase *E. coli* at time 15 min in MM + G, and stationary phase *E. coli* at time 15 min in MM + G exposed to 4 µg mL⁻¹ PmB. The peak at ~1800 *m/z* corresponds to native hexa-acyl diphosphoryl lipid

A containing four C14:0 3-OH, one C14:0 and one C12:0[100]. The peaks at ~1710 and ~1800 *m/z* correspond to the subtraction or addition of phosphate to the native form of lipid A. **b**, Overlaid mass spectra of stationary phase *E. coli* at time 15 min in MM + G (black) and stationary phase *E. coli* at time 15 min in MM + G exposed to 4 µg mL⁻¹ PmB (red), indicating a substantial reduction in native lipid A.

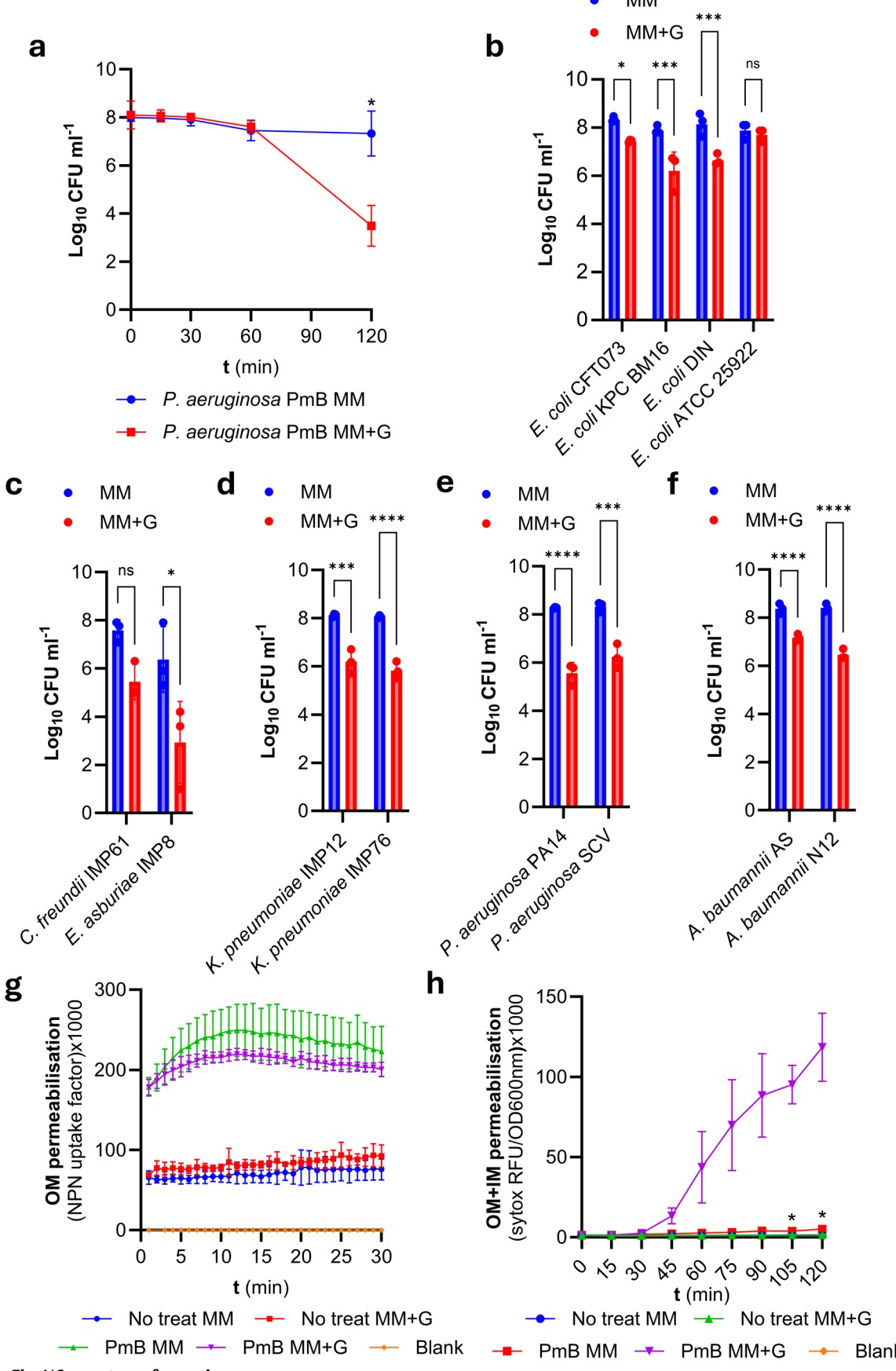

**Extended Data Fig. 4 | See next page for caption.**

**Extended Data Fig. 4 | Tolerance to PmB is conserved under nutrient-limiting conditions. a**. Survival of stationary phase *Pseudomonas aeruginosa* PA14 exposed to 4 μg ml$^{-1}$ PmB in MM ± glucose, as determined by CFU counts. **b**, Survival of a panel of *E. coli* clinical isolates cultured to stationary phase, exposed to 4 μg ml$^{-1}$ PmB in MM ± glucose for 2 h. **c**, Survival of *C. freundii* and *E. asburiase* clinical isolates cultured to stationary phase, exposed to 4 μg ml$^{-1}$ PmB in MM ± glucose for 2 h. **d**, Survival of a *K. pneumoniae* clinical isolates cultured to stationary phase, exposed to 4 μg ml$^{-1}$ PmB in MM ± glucose for 2 h. **e**, Survival of *P. aeruginosa* clinical isolates cultured to stationary phase, exposed to 4 μg ml$^{-1}$ PmB in MM ± glucose for 2 h. **f**, Survival of *A. baumannii* clinical

isolates cultured to stationary phase, exposed to 4 μg ml$^{-1}$ PmB. **g**, OM disruption of stationary phase *P. aeruginosa* cells during the first 20 min of exposure to 4 μg ml$^{-1}$ PmB, as determined by uptake of the NPN fluorescent dye. **h**, OM and IM disruption of stationary phase *P. aeruginosa* exposed to 4 μg ml$^{-1}$ PmB in MM ± glucose, as determined by uptake of the fluorescent dye SYTOX green. For **g**, and **h**, the blank value refers to the relevant fluorophore in medium without bacteria. All experiments were replicated in n = 3 independent assays, error bars show the standard deviation of the mean. Significant differences were determined by two-way (**a - h**) repeated measures ANOVA. P= *<0.05, **<0.01, ***<0.001, ****<0.0001, ns=not significant.

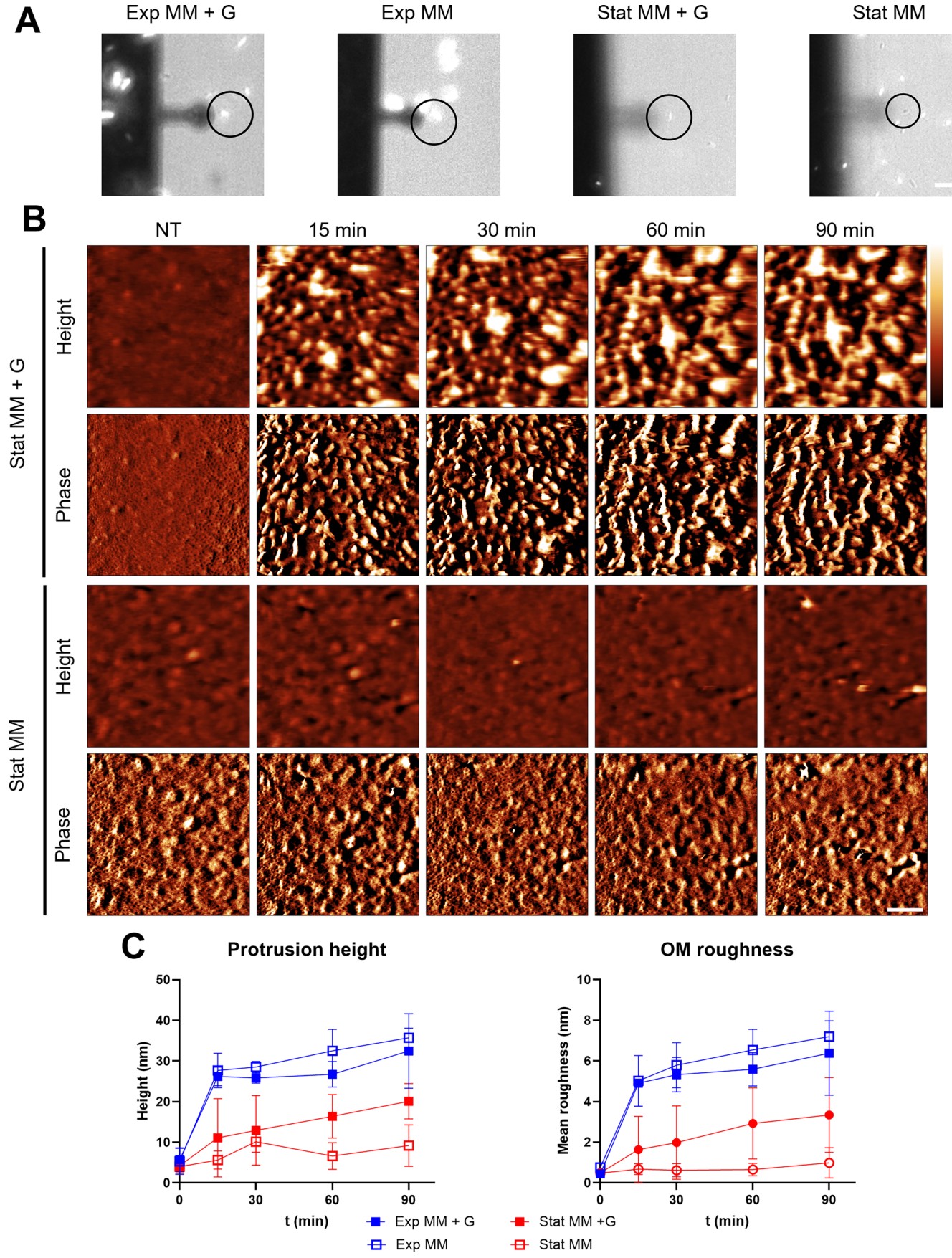

**Extended Data Fig. 5 | See next page for caption.**

**Extended Data Fig. 5 | PmB leads to morphological changes to the OM at the nanoscale. a**, Combined brightfield and fluorescence (SYTOX) images of the AFM scan region for the experiments in Fig. 1b with exponential (Exp) and stationary (Stat) phase *E. coli* at 90 min post PmB treatment (the circled cells are those chosen for the image sequences shown here and were all SYTOX-positive by 90 mins, except for stationary phase *E. coli* in MM). **b**, Higher-magnification AFM height and phase scans of stationary phase *E. coli*. The surface of untreated cells is covered in a network of pores, here best visible in the phase images. The resolution of pores (trimeric OMPs) is ultimately compromised by the progressive roughening of the OM after PmB is added in the presence of

glucose (MM + G). In MM + G, PmB caused disruption to the OM in the form of protrusions. Without glucose the appearance of the OM at the nanoscale did not change. Scalebars: **a**, 4 μm, **b**, 100 nm; height scales (scale inset in first row of image at t = 90 min): 20 nm; phase scale: 2 deg (row 2) and 1 deg (row 4). **c**, Quantification of protrusion height (left) and OM mean roughness (right). Protrusion height data are presented as the mean ± SD of median values of the height of features above 50% of the maximum height of each image. Mean roughness values were computed by Gwyddion and data are presented as the mean ± SD. Measurements were taken from 3 different 500 nm scans of 3 different *E. coli* MG1655 cells imaged in separate experiments.

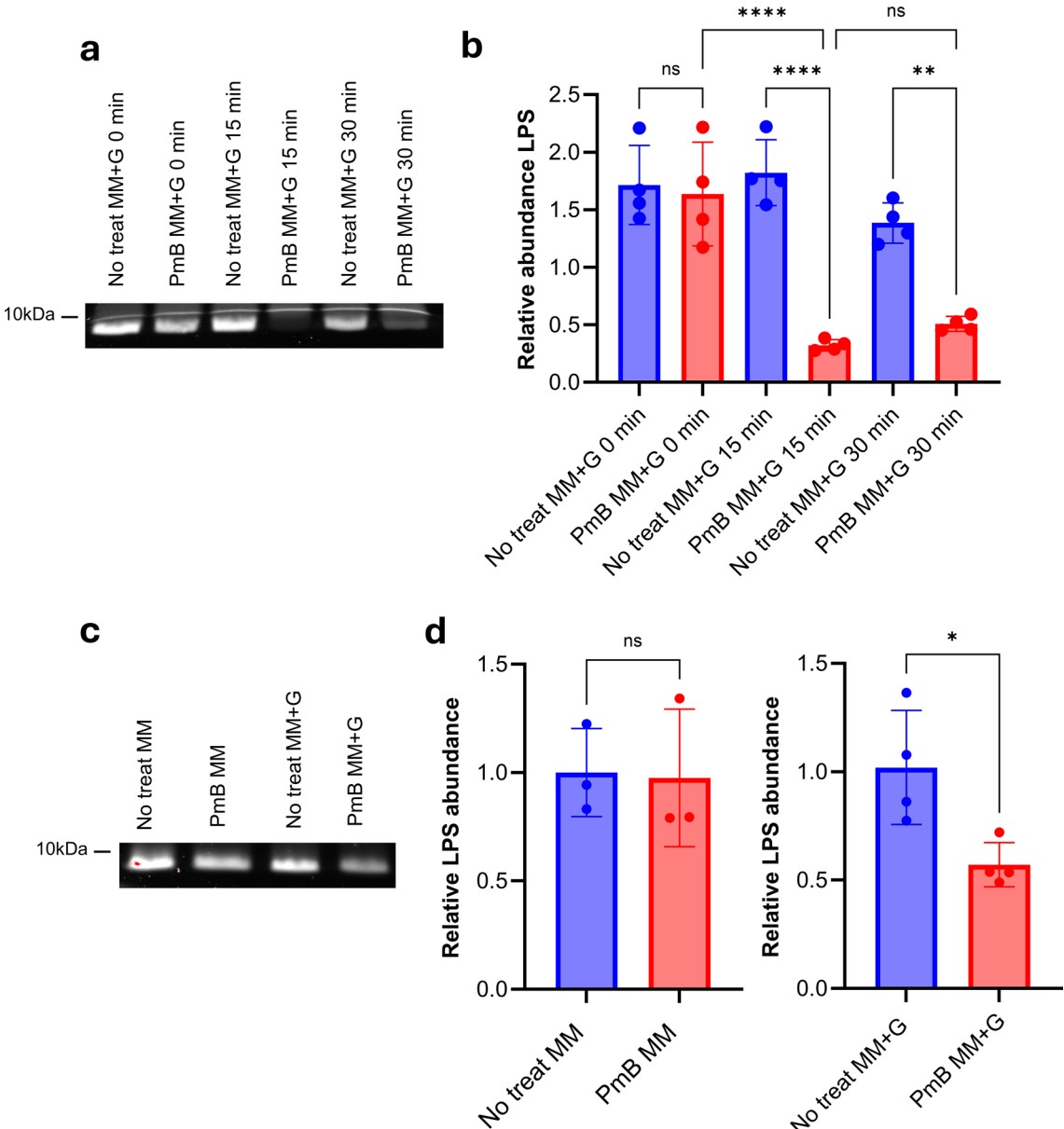

**Extended Data Fig. 6 | PmB-induced LPS loss is maximal at 15 minutes and is a conserved phenotype. a**. Levels of total LPS in *E. coli* exposed, or not, to 4 µg ml⁻¹ PmB for 0-, 15-, and 30 min in MM + G. **b**, Bar graph of LPS levels according to densitometry analysis of **a**. Densitometric values were interpolated according to the standard curve in Supplementary Fig. S7C and subsequently normalized to the No treatment controls, to give 'Relative abundance LPS'. **c**, Levels of total LPS in *P. aeruginosa* exposed, or not, to 4 µg ml⁻¹ PmB in MM ± glucose. **d**, Bar graph of LPS levels according to densitometry analysis of **a**. Experiments were replicated in n = 4 **a**, **b** or n = 3 **c**, **d** independent assays. Error bars show the standard deviation of the mean. Significant differences were determined by one-way ANOVA P= **\*\*<0.01, \*\*\*\*<0.0001, ns=not significant.

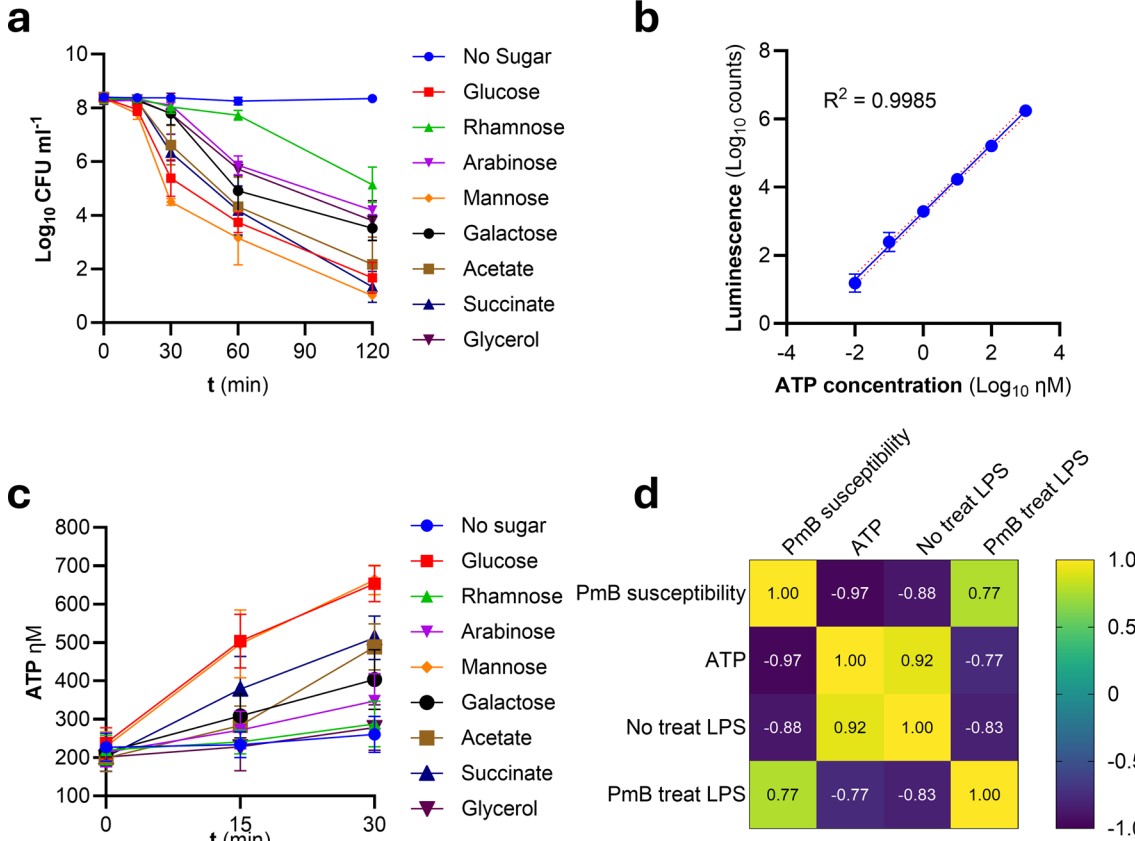

**Extended Data Fig. 7 | PmB killing requires ATP. a**, Survival of stationary phase *E. coli* exposed to 4 µg ml⁻¹ PmB in MM ± equimolar concentrations of different sugars, as determined by CFU counts. **b**, Standard curve plotting Log₁₀ ATP concentration (nM) against Log₁₀ luminescence counts. Simple linear regression was performed using prism version 10.4.1, where the blue line represents the line of best fit, and the red dotted line the 95% confidence interval.

**c**, ATP concentration according to standard curve interpolation, of stationary phase *E. coli* incubated in equimolar concentrations of different sugars across a 30-minute time course. **d**, Four-way correlation matrix showing correlation coefficients of PmB susceptibility, ATP concentration and densitometric analysis pre- and post-PmB treatment. All experiments were replicated in n = 3 independent assays. Error bars show the standard deviation of the mean.

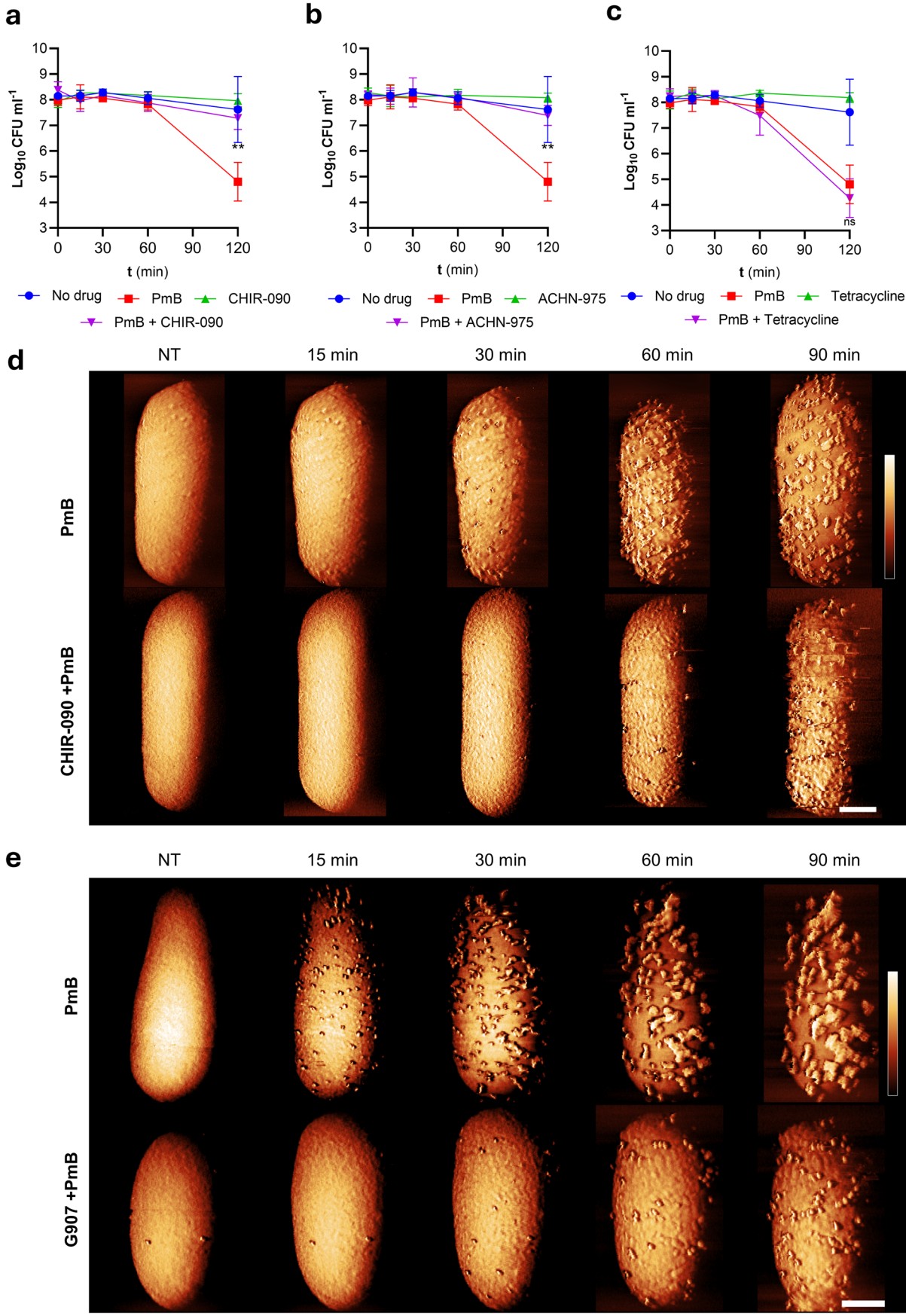

**Extended Data Fig. 8 | See next page for caption.**

**Extended Data Fig. 8 | PmB killing requires LPS synthesis. a, b, c** Survival of stationary phase *P. aeruginosa* exposed, or not, to 4 µg ml⁻¹ PmB in MM + G with or without 1X MIC of LpxC inhibitors CHIR-090 (**a**) or ACHN-975 (**b**) or the protein synthesis inhibitor tetracycline (**c**). (**d**) AFM phase images showing stationary phase *E. coli* MG1655 cells exposed to 2.5 µg ml⁻¹ PmB in the presence or not of 0.125 µg ml⁻¹ CHIR-090 in MM + G, shown as a function of time. (**e**) AFM phase images showing stationary phase *E. coli imp* 4231 cells exposed to 2.5 µg ml⁻¹ PmB in the presence or not of 0.5 µg ml⁻¹ G907 in MM + G, shown as a function of time. Scalebar: **d**, **e** 250 nm. Phase scale (scale inset in first row of **d** and **e**, at t = 90 min): **d**, 2.5 deg, **e** 4 deg (top), 6 deg (bottom) All experiments were replicated in n = 3 independent assays. Error bars show the standard deviation of the mean. Significant differences were determined by two-way repeated measures ANOVA between PmB-treated and PmB + antibiotic-treated conditions. P= **<0.01, ns=not significant.

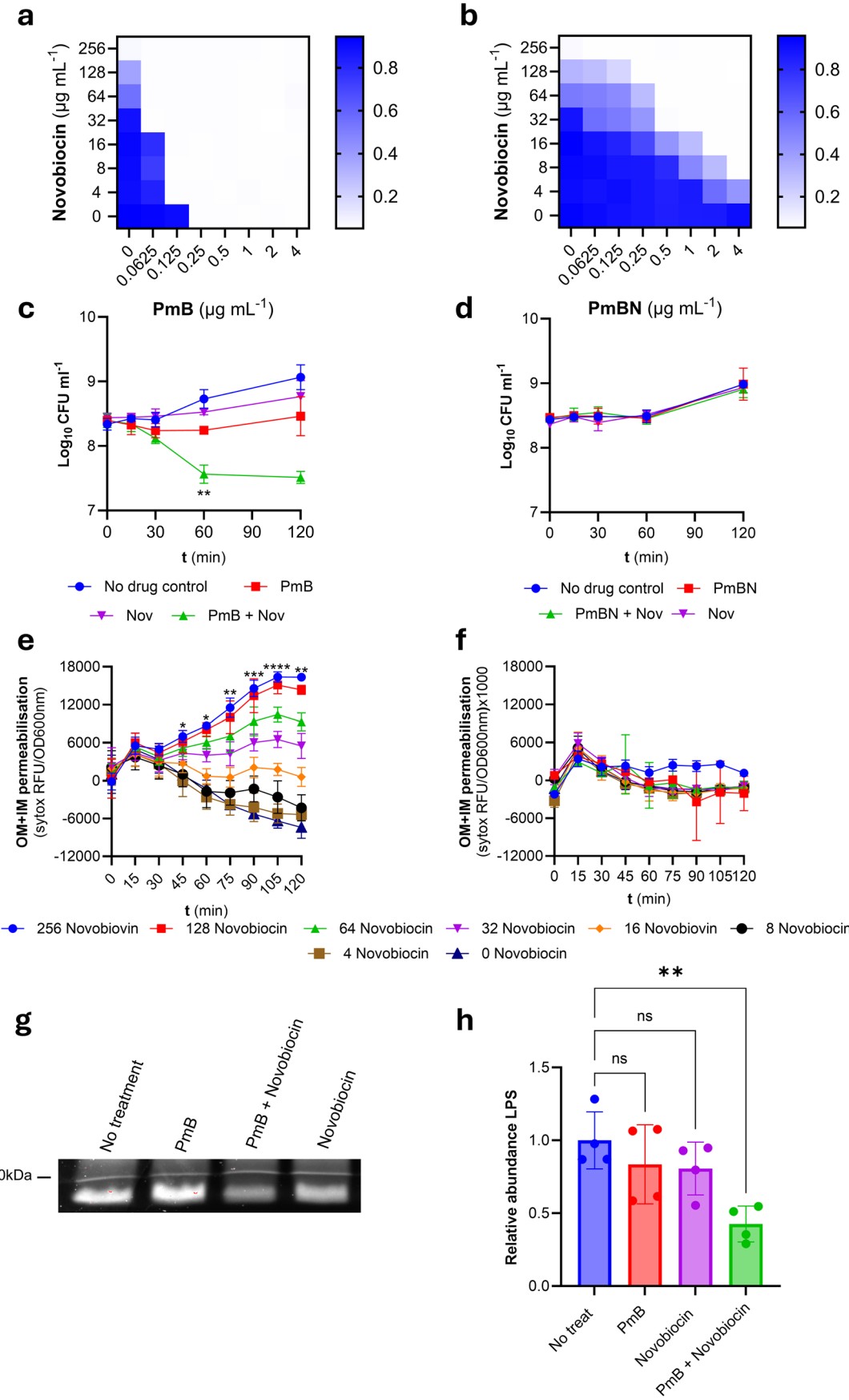

**Extended Data Fig. 9 | See next page for caption.**

**Extended Data Fig. 9 | Novobiocin promotes PmB-mediated LPS loss and killing.** Previous work has shown that the antibiotic novobiocin promotes the rate of LPS transport from IM to OM via an interaction with LptB[41]. In turn, this leads to increased susceptibility to polymyxin B, although the mechanism was not established[42]. Based on the findings described in this manuscript, we hypothesised that increased LPS transport to the OM would promote PmB-mediated LPS loss, which correlates with bacterial killing by the antibiotic. **a, b**, Checkerboard broth microdilution assay showing the synergistic growth-inhibitory interaction between novobiocin and PmB (**a**), or PmBN (**b**) against *E. coli*, as determined OD$_{595nm}$ after 18 hr incubation and in line with previous findings[42]. Note: PmBN is an inactive polymyxin analogue that served as a useful control for increased entry of novobiocin caused by OM disruption. (**c, d**) Survival of *E. coli* exposed, or not, to 1 μg ml$^{-1}$ PmB (**c**) or 1 μg ml$^{-1}$ PmBN (**d**) with or without 32 μg ml$^{-1}$ novobiocin. These concentrations were chosen because they showed maximal synergy in the PmBN checkerboard assay. For (**c**), there was no killing of *E. coli* by the polymyxin or novobiocin alone, but a significant reduction in the viability of *E. coli* in the presence of both antibiotics. By contrast, there was no reduction in bacterial viability in the presence of PmBN with novobiocin. Therefore, bacterial killing in these assays appears to be PmB-mediated, rather than increased ingress of novobiocin into cells caused by OM disruption. **e, f**, Combined OM and IM disruption of stationary phase

*E. coli* exposed to 1 μg ml$^{-1}$ PmB (**e**) or 1 μg ml$^{-1}$ PmBN (**f**) in MHB with and without 32 μg ml$^{-1}$ novobiocin, as determined by uptake of the fluorescent dye SYTOX green. Crucially, in the presence of PmBN, novobiocin did not cause membrane disruption, even at concentrations well above those required to inhibit growth in the checkerboard assay. However, novobiocin promoted PmB-mediated membrane disruption in a dose-responsive manner, confirming that novobiocin promotes the activity of PmB by increasing IM disruption, which is the key step for lethality[27,42]. Next, we wanted to examine the impact of novobiocin on PmB-mediated LPS loss. (**g**) Representative SDS-PAGE image of LPS band intensity of stationary phase *E. coli* exposed, or not, to 1 μg ml$^{-1}$ PmB with and without 32 μg/ml novobiocin for 15 min in MHB. (**h**) Densitometric analysis of the LPS gel in (**g**), showing that PmB or novobiocin exposure alone at 1 μg ml$^{-1}$ or 32 μg ml$^{-1}$ respectively, had minimal impact on LPS abundance after 15 mins incubation (N = 4). However, there was a significant drop in LPS levels when PmB and novobiocin were used in combination at these concentrations. Therefore, we concluded that novobiocin enhances PmB activity by promoting LPS loss, most likely via its reported effect on increasing LPS transport from the IM to OM[41,42]. Unless otherwise stated, all experiments were replicated in n = 3 independent assays. Error bars show the standard deviation of the mean. Significant differences were determined by one- (**h**) or two-way repeated measures ANOVA (**c, d, e, f**). P= *<0.05, **<0.01, ***<0.001, ****<0.0001, ns=not significant.

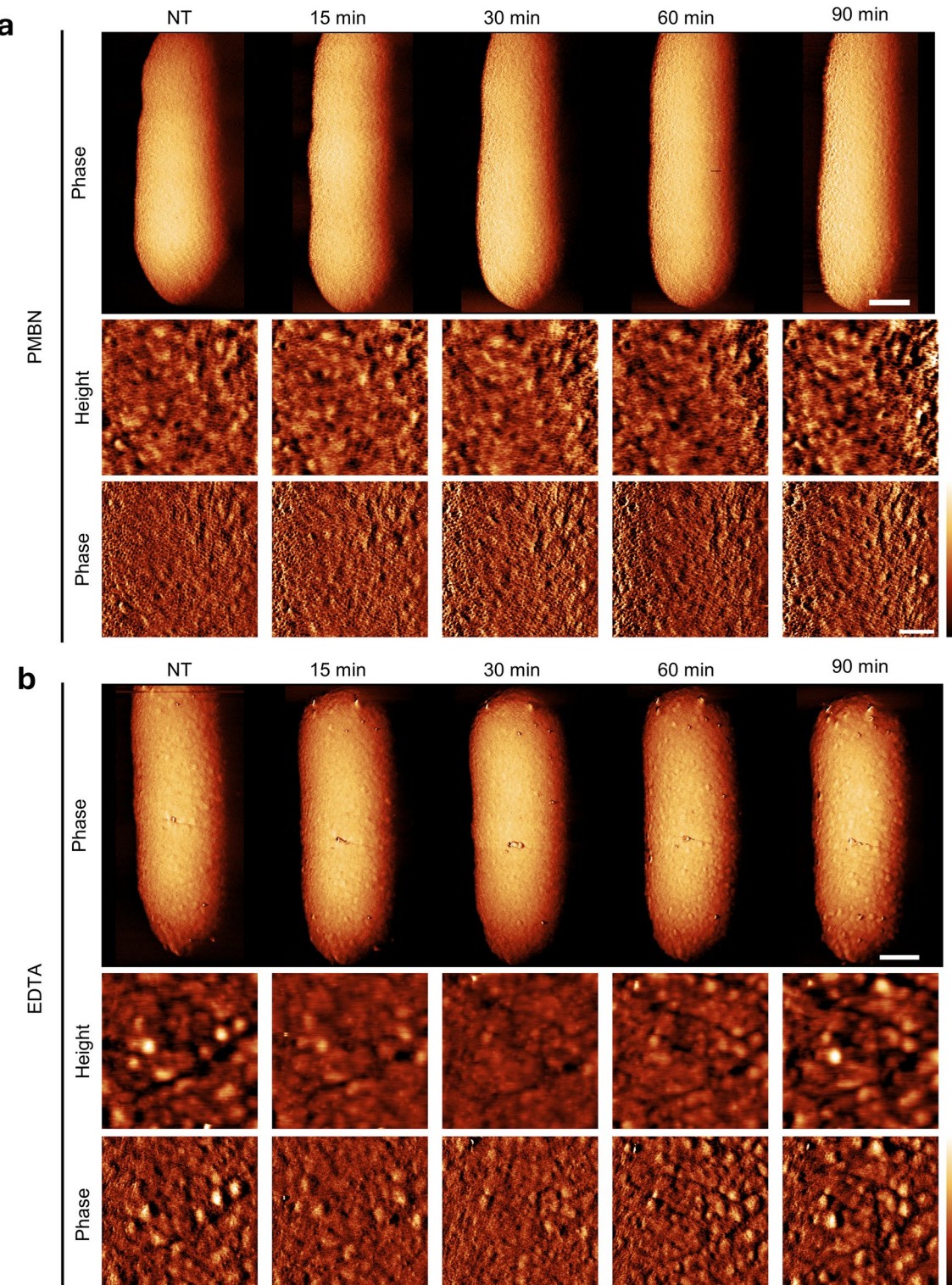

**Extended Data Fig. 10 | Low-dose PmBN and EDTA exposure has little effect on the OM at the nanoscale.** AFM low and high magnification scans showing stationary phase *E. coli* MG1655 exposed to 2.5 μg ml⁻¹ PMBN (**a**) or 10 mM EDTA (**b**) in the presence of glucose for up to 90 minutes. Scalebars: large scans 250 nm, higher-magnification scans 100 nm; phase scale (scale inset in third row of **a** and **b**, at t = 90 min) (from top to bottom): **a**, 4 deg (row 1) and 0.7 deg (row 3), **b**, 5 deg (row 1) and 1 deg (row 3). Height scale (scale inset in third row of **a** and **b**, at t = 90 min): **a**, 3 nm, **b**, 10 nm. (representative images shown of N = 1).

# Reporting Summary

## Statistics

For all statistical analyses, confirm that the following items are present in the figure legend, table legend, main text, or Methods section.

| n/a | Confirmed | |
|---|---|---|
| ☐ | ☒ | The exact sample size (*n*) for each experimental group/condition, given as a discrete number and unit of measurement |
| ☐ | ☒ | A statement on whether measurements were taken from distinct samples or whether the same sample was measured repeatedly |
| ☐ | ☒ | The statistical test(s) used AND whether they are one- or two-sided<br>*Only common tests should be described solely by name; describe more complex techniques in the Methods section.* |
| ☒ | ☐ | A description of all covariates tested |
| ☐ | ☒ | A description of any assumptions or corrections, such as tests of normality and adjustment for multiple comparisons |
| ☐ | ☒ | A full description of the statistical parameters including central tendency (e.g. means) or other basic estimates (e.g. regression coefficient) AND variation (e.g. standard deviation) or associated estimates of uncertainty (e.g. confidence intervals) |
| ☐ | ☒ | For null hypothesis testing, the test statistic (e.g. $F$, $t$, $r$) with confidence intervals, effect sizes, degrees of freedom and $P$ value noted<br>*Give P values as exact values whenever suitable.* |
| ☒ | ☐ | For Bayesian analysis, information on the choice of priors and Markov chain Monte Carlo settings |
| ☒ | ☐ | For hierarchical and complex designs, identification of the appropriate level for tests and full reporting of outcomes |
| ☒ | ☐ | Estimates of effect sizes (e.g. Cohen's *d*, Pearson's *r*), indicating how they were calculated |

*Our web collection on statistics for biologists contains articles on many of the points above.*

## Software and code

Policy information about availability of computer code

| Data collection | ImageJ (1.54f) software was used for densitometry analyses of SDS-PAGE and western blots. |
|---|---|
| Data analysis | All statistical analyses were performed using GraphPad Prism 7 software (GraphPad Software Inc, USA). All AFM image analyses were performed with Gwyddion 2.65. |

For manuscripts utilizing custom algorithms or software that are central to the research but not yet described in published literature, software must be made available to editors and reviewers. We strongly encourage code deposition in a community repository (e.g. GitHub). See the Nature Portfolio guidelines for submitting code & software for further information.

## Data

Policy information about availability of data

All manuscripts must include a data availability statement. This statement should provide the following information, where applicable:
- Accession codes, unique identifiers, or web links for publicly available datasets
- A description of any restrictions on data availability
- For clinical datasets or third party data, please ensure that the statement adheres to our policy

All data supporting the conclusions of this work can be accessed from the UCL Research Data Repository: https://doi.org/10.5522/04/29282072.v1. With the exception of AFM metadata, all data are also provided as source data in this manuscript.

# Research involving human participants, their data, or biological material

Policy information about studies with human participants or human data. See also policy information about sex, gender (identity/presentation), and sexual orientation and race, ethnicity and racism.

| | |
|---|---|
| Reporting on sex and gender | N/A |
| Reporting on race, ethnicity, or other socially relevant groupings | N/A |
| Population characteristics | N/A |
| Recruitment | N/A |
| Ethics oversight | N/A |

Note that full information on the approval of the study protocol must also be provided in the manuscript.

# Field-specific reporting

Please select the one below that is the best fit for your research. If you are not sure, read the appropriate sections before making your selection.

☒ Life sciences ☐ Behavioural & social sciences ☐ Ecological, evolutionary & environmental sciences

For a reference copy of the document with all sections, see nature.com/documents/nr-reporting-summary-flat.pdf

# Life sciences study design

All studies must disclose on these points even when the disclosure is negative.

| | |
|---|---|
| Sample size | Sample sizes were based on previous experiments or pilot assays that confirmed the number of replicates used provided appropriate statistical power. Unless stated, all experiments included at least 3 independent assays, which is sufficient for the statistical . |
| Data exclusions | No data were excluded. |
| Replication | Except where stated, all experiments were run using at least 3 independent assays. Details on the number of replicates are provided in figure legends. |
| Randomization | No randomisation was used as measurements were not subjective. |
| Blinding | No blinding was used as measurements were not subjective. |

# Reporting for specific materials, systems and methods

We require information from authors about some types of materials, experimental systems and methods used in many studies. Here, indicate whether each material, system or method listed is relevant to your study. If you are not sure if a list item applies to your research, read the appropriate section before selecting a response.

## Materials & experimental systems

| n/a | Involved in the study |
|---|---|
| ☐ | ☒ Antibodies |
| ☒ | ☐ Eukaryotic cell lines |
| ☒ | ☐ Palaeontology and archaeology |
| ☒ | ☐ Animals and other organisms |
| ☒ | ☐ Clinical data |
| ☒ | ☐ Dual use research of concern |
| ☒ | ☐ Plants |

## Methods

| n/a | Involved in the study |
|---|---|
| ☒ | ☐ ChIP-seq |
| ☒ | ☐ Flow cytometry |
| ☒ | ☐ MRI-based neuroimaging |

## Antibodies

| | |
|---|---|
| Antibodies used | Monoclonal anti-GroEL antibodies were obtained from Abcam (Clone: EPR28718-8; Cat# AB318970 - Lot number 1096886-3) and used at 1:2000 dilution. |

Polyclonal Anti-OmpF antibodies were obtained from Invitrogen (Cat# PAS-121442 - Lot 2K4557550) were used at 1:5000 dilution. Polyclonal Goat Anti-Rabbit HRP conjugated (Cat# AB6721 Lot GR342221) was used at a 1:2000 dilution.

Validation

Validation information can be found on the supplier's website:
Anti-GroEL: https://www.abcam.com/en-us/products/primary-antibodies/groel-antibody-epr28718-8-ab318970?srsltid=AfmBOoqbl6THabrcX38jZGs6cwrK1HtMhjBPB54T9ggNWX7IAbXNFZm1
Anti-OmpF: https://www.thermofisher.com/antibody/product/ompF-Antibody-Polyclonal/PA5-121442
Anti-rabbit HRP: https://www.abcam.com/en-us/products/secondary-antibodies/goat-rabbit-igg-h-l-hrp-ab6721?srsltid=AfmBOopUrB48GUFPwb_wISsCbNA73rqHjmNMH8_wIpp2UaxSQ73F6RZe

## Plants

Seed stocks

N/A

Novel plant genotypes

N/A

Authentication

N/A

