## [Peer Review File · Nature Microbiology]

Polymyxin B lethality requires energy-dependent bacterial outer membrane disruption

Corresponding Author: Dr Andrew Edwards

Version 0:

Reviewer comments:

Reviewer #1

(Remarks to the Author)

This is an interesting and thorough paper that uses a multitude of complementary approaches to better understand the killing action of the antimicrobial polymyxin B. The primary and important finding is that PmB requires metabolic activity to kill, and that the mechanism for this specificity is related to synthesis and turnover of LPS in the outer membrane. The argument in favour of this interpretation of the data is well substantiated through a series of elegant experiments. In particular, using EDTA to show that it is the OM disrupting impact of PmB when there is metabolic activity that is important, while its ability to disrupt the inner membrane and cause death is not glucose metabolism dependent, is very neat. The direct imaging experiments with AFM also convincingly reveal the impact of PmB on OM structure.

There are a number of issues to clarify prior to publication.

1. On p6 it is claimed that loss of LPS was greatest at 15 min and began to recover, but the graph in S8A it states there is no significant difference between 15 min and 30 min. Is there additional data to support the claim on p6? If not, it would be best to remove it.
2. What's the difference between PmB combined with EDTA and PmB-EDTA? Please clarify, as these seem to have a different effect on cell viability (p11).
3. Figure 6 nicely encapsulates the conclusions of the study, but I found the discussion failed to properly describe the figure. Making the ideas more obvious from the discussion would be helpful for the reader.
4. In figure S10D there looks to be a large difference between untreated 15 min and untreated 10 min. If this is actually not significant, it would be good if it could be marked as such on the figure. If it is significant there should be some explanation of how come there is so much variability between the relative abundance of LPS in untreated samples. As the error bar is rather large it might be safer to make the significance comparison with the untreated 0min sample or perhaps to repeat the experiment some more times.
5. Fig S11 B the blue line is claimed to be a linear regression but in reality it appears to be 'join the dots'.

Other minor issues to address:

1. In the introduction there is a tendency to switch between 'glucose', 'sugar' and 'carbohydrate' to avoid repetition of 'glucose'. It is clearer for the reader if 'glucose' is used throughout.
2. In figure 1E and elsewhere one of the data sets is for "blank". This should be explained somewhere in the text.
3. There are a lot of typos in the Atomic force microscopy imaging and analysis section of the methods – e.g. 512 x 51221 pixels, lie instead of line, etc.

Reviewer #2

(Remarks to the Author)

This study closes the loop on how polymyxin B actually kills Gram-negative bacteria while providing a striking example of how to cleverly design an experiment to separate the various effects of a membrane active antibiotic. The work is important as, although the likely need for metabolic activity has been previously suggested, there has been no understanding of how and under which conditions this might be the case and what the implications of such a requirement might be. With the new understanding provided here we can now appreciate that polymyxin B will kill much more slowly when bacteria are in stationary phase and when high concentrations of these poorly tolerated antibiotics cannot be achieved (as is the case when used clinically). By providing mechanistic insight, the study shows how future combinations might be designed and tested that have a greater chance of therapeutic success and a lower risk of resistance development. The study also provides a standout example of how such questions should be approached. In particular the combination of time-resolved microbiological, dye uptake and atomic force microscopy (AFM) data provides a compelling view of how bacterial killing evolves and a powerful means of dissecting the bactericidal mechanism.

I have some very minor comments which the authors may consider if it is felt there is a need to improve the manuscript, though I

expect it can be published as is:

1. In the abstract, is there a need to be explicit that the LPS loss is from the OM?
2. Is there scope to add a perspective sentence to the end of the abstract?
3. It seems a little contrived to frame the study as a hypothesis (last sentence of the abstract). There are plenty of clear hypotheses that are tested in this study, and it would seem to be enough to state that the aim is to provide an unprecedentedly detailed understanding of how polymyxin B functions.
4. The data (Fig. 1D) for mCherry release is less compelling than the other data. Most notably, the well-designed, time-resolved approach provides a key way of linking those processes to which polymyxin lethality can be attributed. Here the data is characterised by a much higher variance than that provided by other methods, and a significant effect is only observed at 90 and 120 mins after the PmB challenge begins. In the same conditions a 2log₁₀ reduction in CFU is already achieved at 30 mins (Fig. 1A). Although the Sytox dye data shows an effect for PmB against stationary phase bacteria in minimal media (which is perhaps not expected from the time-kill data) the timing of the effect of PmB when glucose is added, i.e. at 30 mins connects the Sytox effect and the outcome much better. I wonder if the mCherry release coming at 90 mins is just reflecting death of the cells and provides less insight into mechanism than is asserted?
5. NPN uptake is a very indirect measure of lipid asymmetry or disorder. Can NPN uptake measurements really be used as a proxy for such parameters or should another term be used?
6. The effect of the G907 inhibitor is interesting. Is it possible that it is LPS trafficking rather than synthesis that is hijacked by PmB (trafficking of course follows from synthesis, so this is moot)? Do you see protrusions with this inhibitor or has this not been tested yet?
7. In Supplementary Figure 5B both *A. baumannii* AS and *E. coli* CFT073 appear to have CFU reductions with glucose but are listed as not significant i.e. $p > 0.05$ with ANOVA with one of four possible tests (as described in the statistical analysis). I wonder if a Brown-Forsythe or Welch test would be appropriate here since the CFU data may have unequal variance (even though log transformed) because of different dilutions needed to record higher or lower CFU values?

Reviewer #3

(Remarks to the Author)

It was a pleasure to review the present manuscript titled "Polymyxin B lethality requires energy-dependent outer membrane disruption." This study reports that the lethality of PmB is energy-dependent, and that this is associated with the active transport of LPS to the OM, ultimately leading to blebbing and disruption, allowing exposure of the IM to permeabilizing PmB. The experiments are well controlled, clearly described, and the manuscript is very well written. Although the exact mechanism of LPS shedding is not elucidated in this study, I believe that the observations they have reported reveal a clinically and biologically important paradigm shifting phenomenon that, similar to other classes of antimicrobial drugs, antimicrobial peptides like PmB are only effective against actively growing cells at clinically relevant concentrations. The observations described in this study now make it possible to explore the energy-dependent mechanisms of PmB lethality, which were not previously known to exist. I believe this study is of broad interest to clinical researchers, microbiologists, cell membrane biologists and even bioengineers, and I recommend its publication with very minor revisions. Below I suggest that the authors provide some discussion of the experimental limitations of the study and speculate more deeply on the possible mechanisms of LPS release. To be clear, I do not think any more experiments need to be performed for this study to be published.

Points for additional discussion:

1. It would be interesting to speculate more broadly about the actual mechanism of LPS release (e.g. blebbing, vesiculation, spontaneous shedding) and some questions that could be asked to address this.
2. The role of stress pathways, in particular envelop stress response (e.g. RpoE etc..) might play in responding to or influencing PmB-induced OM remodeling.
3. What effect could PmB remodeling have on the bacterial immune evasion through its effect on OM remodeling?

Decision Letter:

5th June 2025

Dear Andy,

Thank you for your patience while your manuscript "Polymyxin B lethality requires energy-dependent outer membrane disruption" was under peer-review at Nature Microbiology. It has now been seen by 3 referees, whose expertise and comments you will find at the of this email. You will see from their comments below that while they find your work of interest, some important points are raised. We are very interested in the possibility of publishing your study in Nature Microbiology, but would like to consider your response to these concerns in the form of a revised manuscript before we make a final decision on publication.

In particular, you will see that the referees ask for some additional discussion, including on the potential limitations of the study. The rest of the referees' reports are clear and the remaining issues should be straightforward to address.

If you have not done so already please begin to revise your manuscript so that it conforms to our Article format instructions at <http://www.nature.com/nmicrobiol/info/final-submission/>

The usual length limit for a Nature Microbiology Article is six display items (figures or tables) and 4,000 words. We have some flexibility, and can allow a revised manuscript at 4,500 words, but please consider this a firm upper limit. There is a trade-off of ~250 words per display item, so if you need more space, you could move a Figure or Table to Supplementary Information.

Some reduction could be achieved by focusing any introductory material and moving it to the start of your opening 'bold' paragraph, whose function is to outline the background to your work, describe in a sentence your new observations, and explain your main conclusions. The discussion should also be limited. Methods should be described in a separate section following the discussion, we do not place a word limit on Methods.

Nature Microbiology titles should give a sense of the main new findings of a manuscript, and should not contain punctuation. Please keep in mind that we strongly discourage active verbs in titles, and that they should ideally fit within 90 characters each (including spaces).

Please include a data availability statement as a separate section after Methods but before references, under the heading "Data Availability". This section should inform readers about the availability of the data used to support the conclusions of your study. This information includes accession codes to public repositories (data banks for protein, DNA or RNA sequences, microarray, proteomics data etc...), references to source data published alongside the paper, unique identifiers such as URLs to data repository entries, or data set DOIs, and any other statement about data availability. At a minimum, you should include the following statement: "The data that support the findings of this study are available from the corresponding author upon request", mentioning any restrictions on availability. If DOIs are provided, we also strongly encourage including these in the Reference list (authors, title, publisher (repository name), identifier, year). For more guidance on how to write this section please see: <http://www.nature.com/authors/policies/data/data-availability-statements-data-citations.pdf>

To improve the accessibility of your paper to readers from other research areas, please pay particular attention to the wording of the paper's opening bold paragraph, which serves both as an introduction and as a brief, non-technical summary in about 150 words. If, however, you require one or two extra sentences to explain your work clearly, please include them even if the paragraph is over-length as a result. The opening paragraph should not contain references. Because scientists from other sub-disciplines will be interested in your results and their implications, it is important to explain essential but specialised terms concisely. We suggest you show your summary paragraph to colleagues in other fields to uncover any problematic concepts.

If your paper is accepted for publication, we will edit your display items electronically so they conform to our house style and will reproduce clearly in print. If necessary, we will re-size figures to fit single or double column width. If your figures contain several parts, the parts should form a neat rectangle when assembled. Choosing the right electronic format at this stage will speed up the processing of your paper and give the best possible results in print. We would like the figures to be supplied as vector files - EPS, PDF, AI or postscript (PS) file formats (not raster or bitmap files), preferably generated with vector-graphics software (Adobe Illustrator for example). Please try to ensure that all figures are non-flattened and fully editable. All images should be at least 300 dpi resolution (when figures are scaled to approximately the size that they are to be printed at) and in RGB colour format. Please do not submit Jpeg or flattened TIFF files. Please see also 'Guidelines for Electronic Submission of Figures' at the end of this letter for further detail.

Figure legends must provide a brief description of the figure and the symbols used, within 350 words, including definitions of any error bars employed in the figures.

When submitting the revised version of your manuscript, please pay close attention to our [href="https://www.nature.com/nature-research/editorial-policies/image-integrity">Digital Image Integrity Guidelines. and to the following points below:](https://www.nature.com/nature-research/editorial-policies/image-integrity)

EXTENDED DATA FIGURES

Please include a statement before the acknowledgements naming the author to whom correspondence and requests for

materials should be addressed.

Finally, we require authors to include a statement of their individual contributions to the paper -- such as experimental work, project planning, data analysis, etc. -- immediately after the acknowledgements. The statement should be short, and refer to authors by their initials. For details please see the Authorship section of our joint Editorial policies at http://www.nature.com/authors/editorial_policies/authorship.html

* include a point-by-point response to any editorial suggestions and to our referees. Please include your response to the editorial suggestions in your cover letter, and please upload your response to the referees as a separate document.

* ensure it complies with our format requirements for Letters as set out in our guide to authors at www.nature.com/nmicrobiol/info/gta/

* state in a cover note the length of the text, methods and legends; the number of references; number and estimated final size of figures and tables

* resubmit electronically if possible using the link below to access your home page:

Link Redacted

*This url links to your confidential homepage and associated information about manuscripts you may have submitted or be reviewing for us. If you wish to forward this e-mail to co-authors, please delete this link to your homepage first.

Please ensure that all correspondence is marked with your Nature Microbiology reference number in the subject line.

Nature Microbiology is committed to improving transparency in authorship. As part of our efforts in this direction, we are now requesting that all authors identified as 'corresponding author' on published papers create and link their Open Researcher and Contributor Identifier (ORCID) with their account on the Manuscript Tracking System (MTS), prior to acceptance. This applies to primary research papers only. ORCID helps the scientific community achieve unambiguous attribution of all scholarly contributions. You can create and link your ORCID from the home page of the MTS by clicking on 'Modify my Springer Nature account'. For more information please visit www.springernature.com/orcid.

We hope to receive your revised paper within three weeks. If you cannot send it within this time, please let us know.

Yours sincerely,

Reviewer Expertise:

Referee #1: Microscopy, AMF

Referee #2: Antibiotic resistance, membrane biology

Referee #3: Antibiotic resistance, membrane biology

Reviewers Comments:

Reviewer #1 (Remarks to the Author):

This is an interesting and thorough paper that uses a multitude of complementary approaches to better understand the killing action of the antimicrobial polymyxin B. The primary and important finding is that PmB requires metabolic activity to kill, and that the mechanism for this specificity is related to synthesis and turnover of LPS in the outer membrane. The argument in favour of this interpretation of the data is well substantiated through a series of elegant experiments. In particular, using EDTA to show that it is the OM disrupting impact of PmB when there is metabolic activity that is important, while its ability to disrupt the inner membrane and cause death is not glucose metabolism dependent, is very neat. The direct imaging experiments with AFM also convincingly reveal the impact of PmB on OM structure.

There are a number of issues to clarify prior to publication.

1. On p6 it is claimed that loss of LPS was greatest at 15 min and began to recover, but the graph in S8A it states there is no significant difference between 15 min and 30 min. Is there additional data to support the claim on p6? If not, it would be best to remove it.
2. What's the difference between PmB combined with EDTA and PmB-EDTA? Please clarify, as these seem to have a different effect on cell viability (p11).
3. Figure 6 nicely encapsulates the conclusions of the study, but I found the discussion failed to properly describe the figure. Making the ideas more obvious from the discussion would be helpful for the reader.

4. In figure S10D there looks to be a large difference between untreated 15 min and untreated 10 min. If this is actually not significant, it would be good if it could be marked as such on the figure. If it is significant there should be some explanation of how come there is so much variability between the relative abundance of LPS in untreated samples. As the error bar is rather large it might be safer to make the significance comparison with the untreated 0min sample or perhaps to repeat the experiment some more times.
5. Fig S11 B the blue line is claimed to be a linear regression but in reality it appears to be 'join the dots'.

Other minor issues to address:

1. In the introduction there is a tendency to switch between 'glucose', 'sugar' and 'carbohydrate' to avoid repetition of 'glucose'. It is clearer for the reader if 'glucose' is used throughout.
2. In figure 1E and elsewhere one of the data sets is for "blank". This should be explained somewhere in the text.
3. There are a lot of typos in the Atomic force microscopy imaging and analysis section of the methods – e.g. 512 x 51221 pixels, lie instead of line, etc.

Reviewer #2 (Remarks to the Author):

This study closes the loop on how polymyxin B actually kills Gram-negative bacteria while providing a striking example of how to cleverly design an experiment to separate the various effects of a membrane active antibiotic. The work is important as, although the likely need for metabolic activity has been previously suggested, there has been no understanding of how and under which conditions this might be the case and what the implications of such a requirement might be. With the new understanding provided here we can now appreciate that polymyxin B will kill much more slowly when bacteria are in stationary phase and when high concentrations of these poorly tolerated antibiotics cannot be achieved (as is the case when used clinically). By providing mechanistic insight, the study shows how future combinations might be designed and tested that have a greater chance of therapeutic success and a lower risk of resistance development. The study also provides a standout example of how such questions should be approached. In particular the combination of time-resolved microbiological, dye uptake and atomic force microscopy (AFM) data provides a compelling view of how bacterial killing evolves and a powerful means of dissecting the bactericidal mechanism.

I have some very minor comments which the authors may consider if it is felt there is a need to improve the manuscript, though I expect it can be published as is:

1. In the abstract, is there a need to be explicit that the LPS loss is from the OM?
2. Is there scope to add a perspective sentence to the end of the abstract?
3. It seems a little contrived to frame the study as a hypothesis (last sentence of the abstract). There are plenty of clear hypotheses that are tested in this study, and it would seem to be enough to state that the aim is to provide an unprecedentedly detailed understanding of how polymyxin B functions.
4. The data (Fig. 1D) for mCherry release is less compelling than the other data. Most notably, the well-designed, time-resolved approach provides a key way of linking those processes to which polymyxin lethality can be attributed. Here the data is characterised by a much higher variance than that provided by other methods, and a significant effect is only observed at 90 and 120 mins after the PmB challenge begins. In the same conditions a 2log₁₀ reduction in CFU is already achieved at 30 mins (Fig. 1A). Although the Sytox dye data shows an effect for PmB against stationary phase bacteria in minimal media (which is perhaps not expected from the time-kill data) the timing of the effect of PmB when glucose is added, i.e. at 30 mins connects the Sytox effect and the outcome much better. I wonder if the mCherry release coming at 90 mins is just reflecting death of the cells and provides less insight into mechanism than is asserted?
5. NPN uptake is a very indirect measure of lipid asymmetry or disorder. Can NPN uptake measurements really be used as a proxy for such parameters or should another term be used?
6. The effect of the G907 inhibitor is interesting. Is it possible that it is LPS trafficking rather than synthesis that is hijacked by PmB (trafficking of course follows from synthesis, so this is moot)? Do you see protrusions with this inhibitor or has this not been tested yet?
7. In Supplementary Figure 5B both *A. baumannii* AS and *E. coli* CFT073 appear to have CFU reductions with glucose but are listed as not significant i.e. $p > 0.05$ with ANOVA with one of four possible tests (as described in the statistical analysis). I wonder if a Brown-Forsythe or Welch test would be appropriate here since the CFU data may have unequal variance (even though log transformed) because of different dilutions needed to record higher or lower CFU values?

Reviewer #3 (Remarks to the Author):

It was a pleasure to review the present manuscript titled "Polymyxin B lethality requires energy-dependent outer membrane disruption." This study reports that the lethality of PmB is energy-dependent, and that this is associated with the active transport of LPS to the OM, ultimately leading to blebbing and disruption, allowing exposure of the IM to permeabilizing PmB. The experiments are well controlled, clearly described, and the manuscript is very well written. Although the exact mechanism of LPS shedding is not elucidated in this study, I believe that the observations they have reported reveal a clinically and biologically important paradigm shifting phenomenon that, similar to other classes of antimicrobial drugs, antimicrobial peptides like PmB are only effective against actively growing cells at clinically relevant concentrations. The observations described in this study now make it possible to explore the energy-dependent mechanisms of PmB lethality, which were not previously known to exist. I believe this study is of broad interest to clinical researchers, microbiologists, cell membrane biologists and even bioengineers, and I recommend its publication with very minor revisions. Below I suggest that the authors provide some discussion of the experimental limitations of the study and speculate more deeply on the possible mechanisms of LPS release. To be clear, I do not think any more experiments need to be performed for this study to be published.

Points for additional discussion:

1. It would be interesting to speculate more broadly about the actual mechanism of LPS release (e.g. blebbing, vesiculation, spontaneous shedding) and some questions that could be asked to address this.
2. The role of stress pathways, in particular envelope stress response (e.g. RpoE etc..) might play in responding to or influencing PmB-induced OM remodeling.
3. What effect could PmB remodeling have on the bacterial immune evasion through its effect on OM remodeling?

Version 1:

Reviewer comments:

Reviewer #1

(Remarks to the Author)

I am happy with the manuscript as is.

One outstanding correction - it still reads "lie" where it should be "line" on line 556.

Reviewer #3

(Remarks to the Author)

The authors have addressed all of my concerns.

Decision Letter:

16th July 2025

Dear Professor Edwards,

Thank you for your patience while your manuscript "Polymyxin B lethality requires energy-dependent outer membrane disruption" was under peer review at Nature Microbiology. It has now been seen by our referees, and in the light of their advice I am delighted to say that we can in principle offer to publish it. Before I can send you the accept-in-principle decision letter, however, we will need you to convert 10 of the Supplementary Information figures to Extended Data Figures and change the text accordingly (e.g., Suppl. Fig 1 should become Extended Data Fig 1 etc). Please upload all main and extended data figures separately.

Please resubmit electronically by July 24:

* the final version of the text (not including the figures) in either Word or Latex.

* publication-quality figures. For more details, please refer to our Figure Guidelines, which is available here: https://mts-nmicrobiol.nature.com/letters/Figure_guidelines.pdf

* Extended Data & Supplementary Information, as instructed

Please use the following link to access your home page:

Link Redacted

* This url links to your confidential homepage and associated information about manuscripts you may have submitted or be reviewing for us. If you wish to forward this e-mail to co-authors, please delete this link to your homepage first.

ORCID

Nature Microbiology is committed to improving transparency in authorship. As part of our efforts in this direction, we are now requesting that all authors identified as 'corresponding author' create and link their Open Researcher and Contributor Identifier (ORCID) with their account on the Manuscript Tracking System (MTS) prior to acceptance. ORCID helps the scientific community achieve unambiguous attribution of all scholarly contributions. For more information please visit <http://www.springernature.com/orcid>

For all corresponding authors listed on the manuscript, please follow the instructions in the link below to link your ORCID to your

account on our MTS before submitting the final version of the manuscript. If you do not yet have an ORCID you will be able to create one in minutes.

IMPORTANT: All authors identified as 'corresponding author' on the manuscript must follow these instructions. Non-corresponding authors do not have to link their ORCIDs but are encouraged to do so. Please note that it will not be possible to add/modify ORCIDs at proof. Thus, if they wish to have their ORCID added to the paper they must also follow the above procedure prior to acceptance.

To support ORCID's aims, we only allow a single ORCID identifier to be attached to one account. If you have any issues attaching an ORCID identifier to your MTS account, please contact the [Platform Support Helpdesk](http://platformsupport.nature.com/).

Yours sincerely,

Reviewer Comments:

Reviewer #1 (Remarks to the Author):

I am happy with the manuscript as is.
One outstanding correction - it still reads "lie" where it should be "line" on line 556.

Reviewer #3 (Remarks to the Author):

The authors have addressed all of my concerns.

Version 2:

Decision Letter:

Our ref: NMICROBIOL-25041411B

30th July 2025

Dear Andy,

Thank you for submitting your revised manuscript "Polymyxin B lethality requires energy-dependent outer membrane disruption" (NMICROBIOL-25041411B). It has now been seen by the original referees and their comments are below. The reviewers find that the paper has improved in revision, and therefore we'll be happy in principle to publish it in Nature Microbiology, pending minor revisions to comply with our editorial and formatting guidelines.

We are now performing detailed checks on your paper and will send you a checklist detailing our editorial and formatting requirements in about two weeks. Please do not upload the final materials and make any revisions until you receive this additional information from us.

Thank you again for your interest in Nature Microbiology. Please do not hesitate to contact me if you have any questions.

Sincerely,

Version 3:

Decision Letter:

26th August 2025

Dear Dr Edwards,

I am pleased to accept your Article "Polymyxin B lethality requires energy-dependent bacterial outer membrane disruption" for publication in Nature Microbiology. Thank you for having chosen to submit your work to us and many congratulations.

Authors may need to take specific actions to achieve compliance with funder and institutional open access mandates. If your research is supported by a funder that requires immediate open access (e.g. according to [a Plan S principles](https://www.springernature.com/gp/open-science/plan-s-compliance) or the [NIH public access policy](https://www.springernature.com/gp/open-science/us-federal-agency-compliance)) then you should select the gold OA route, and we will direct you to the compliant route where possible. Because authors warrant under our subscription licensing terms that they haven't committed to licensing any version of their article under a licence inconsistent with the terms of our agreement – including the applicable embargo period – publication under the subscription model isn't suitable for authors whose funders require no embargo.

With kind regards,

P.S. Click on the following link if you would like to recommend Nature Microbiology to your librarian
<http://www.nature.com/subscriptions/recommend.html#forms>

** Visit the Springer Nature Editorial and Publishing website at http://editorial-jobs.springernature.com?utm_source=ejP_NMicro_email&utm_medium=ejP_NMicro_email&utm_campaign=ejp_NMicro for more information about our career opportunities. If you have any questions please click [here](mailto:editorial.publishing.jobs@springernature.com).

NMICROBIOL-25041411 Response to reviewers' comments

Reviewer comments in black.

Our responses in Blue.

We would like to begin by thanking the three reviewers for their time, effort, and thoughtful feedback, as well as for their strong support of our work. We have carefully addressed each of the points raised below, within the constraints of the word limit, and we believe these revisions have significantly improved the manuscript.

Reviewer #1 (Remarks to the Author):

This is an interesting and thorough paper that uses a multitude of complementary approaches to better understand the killing action of the antimicrobial polymyxin B. The primary and important finding is that PmB requires metabolic activity to kill, and that the mechanism for this specificity is related to synthesis and turnover of LPS in the outer membrane. The argument in favour of this interpretation of the data is well substantiated through a series of elegant experiments. In particular, using EDTA to show that it is the OM disrupting impact of PmB when there is metabolic activity that is important, while its ability to disrupt the inner membrane and cause death is not glucose metabolism dependent, is very neat. The direct imaging experiments with AFM also convincingly reveal the impact of PmB on OM structure.

There are a number of issues to clarify prior to publication.

1. On p6 it is claimed that loss of LPS was greatest at 15 min and began to recover, but the graph in S8A it states there is no significant difference between 15 min and 30 min. Is there additional data to support the claim on p6? If not, it would be best to remove it.

This claim has been removed as suggested. Please see lines: 160-161, "Analysis of bacterial cells found a corresponding loss of LPS from *E. coli* exposed to PmB in the presence but not absence of glucose (Fig. 2C,D, Supplementary Fig S7), [removed claim] which...".

2. What's the difference between PmB combined with EDTA and PmB-EDTA? Please clarify, as these seem to have a different effect on cell viability (p11).

This question refers to Fig. 4, where the data in Fig. 4D come from experiments where bacteria were exposed to each agent (PmB, PmBN or EDTA) or none, for 15 mins, and then washed before incubation in the presence of the indicated concentration of PmB in MM+G for 2 h. For figures 4F and 4G, bacteria were incubated in the presence of PmB, EDTA or both over 2h in MM or MM+G.

When *E. coli* was pretreated with a membrane disrupting agent before washing and subsequent exposure to PmB (e.g. Figs. 4A, 4D), there is a greater level of bacterial killing / lower cell viability when compared with experiments without a pre-treatment step (e.g. Figs. 3C, 3D, 3F). A possible explanation is that released LPS sequesters PmB, reducing its activity [Yokota et al., 2018 IJAA]. Since this is washed away in the pre-treatment experiments, it cannot reduce the activity of the subsequent PmB dose. Furthermore, once the OM is compromised by the membrane disrupting pre-

treatment, subsequent PmB treatment can efficiently access the IM, leading to bacterial killing. This has now been clarified in the manuscript (please see lines: 244, 258-259, 263-270).

3. Figure 6 nicely encapsulates the conclusions of the study, but I found the discussion failed to properly describe the figure. Making the ideas more obvious from the discussion would be helpful for the reader.

We have revised the discussion to provide a clearer overview of the key conclusions, as well as areas that require further work and how this could be done (please see lines 393, 396-404, second and third paragraph of the Discussion).

4. In figure S10D there looks to be a large difference between untreated 15 min and untreated 10 min. If this is actually not significant, it would be good if it could be marked as such on the figure. If it is significant there should be some explanation of how come there is so much variability between the relative abundance of LPS in untreated samples. As the error bar is rather large it might be safer to make the significance comparison with the untreated 0min sample or perhaps to repeat the experiment some more times.

We thank the reviewer for this suggestion and the data have been re-analysed as suggested. There is no significant difference between untreated 0 min and untreated 15 min and this is now indicated on the graph. Variability is likely due to the growth that occurs in MHB, which does not occur in MM+G over the relevant timeframe, resulting in less variability.

5. Fig S11 B the blue line is claimed to be a linear regression but in reality it appears to be 'join the dots'.

We apologise for this mistake, which has now been corrected to plot log ATP against log luminescence and linear regression performed, along with 95% confidence intervals.

Other minor issues to address:

1. In the introduction there is a tendency to switch between 'glucose', 'sugar' and 'carbohydrate' to avoid repetition of 'glucose'. It is clearer for the reader if 'glucose' is used throughout.

We have changed the text as suggested (please see lines: 110, 158).

2. In figure 1E and elsewhere one of the data sets is for "blank". This should be explained somewhere in the text.

We have added explanations of what blank refers to in the legends of Figs. 1 and 5 and Supplementary Figs. S3, S5 and S11 (please see highlighted text of each legend).

3. There are a lot of typos in the Atomic force microscopy imaging and analysis section of the methods – e.g. 512 x 51221 pixels, lie instead of line, etc.

We apologise for the errors. These have been corrected (please see lines: 555-556).

Reviewer #2 (Remarks to the Author):

This study closes the loop on how polymyxin B actually kills Gram-negative bacteria while providing a striking example of how to cleverly design an experiment to separate the various effects of a membrane active antibiotic. The work is important as, although the likely need for metabolic activity has been previously suggested, there has been no understanding of how and under which conditions this might be the case and what the implications of such a requirement might be. With the new understanding provided here we can now appreciate that polymyxin B will kill much more slowly when bacteria are in stationary phase and when high concentrations of these poorly tolerated antibiotics cannot be achieved (as is the case when used clinically). By providing mechanistic insight, the study shows how future combinations might be designed and tested that have a greater chance of therapeutic success and a lower risk of resistance development. The study also provides a standout example of how such questions should be approached. In particular the combination of time-resolved microbiological, dye uptake and atomic force microscopy (AFM) data provides a compelling view of how bacterial killing evolves and a powerful means of dissecting the bactericidal mechanism.

I have some very minor comments which the authors may consider if it is felt there is a need to improve the manuscript, though I expect it can be published as is:

1. In the abstract, is there a need to be explicit that the LPS loss is from the OM?

The abstract has been edited for clarity as suggested (please see lines: 27-28, "LPS loss from the outer membrane (OM), ...").

2. Is there scope to add a perspective sentence to the end of the abstract?

We have added a perspective statement as suggested (please see lines: 31-33). To accommodate this within the word limit, we also changed the abstract slightly to reduce text describing experiments with MCR-1 (please see lines: 28-29).

3. It seems a little contrived to frame the study as a hypothesis (last sentence of the abstract). There are plenty of clear hypotheses that are tested in this study, and it would seem to be enough to state that the aim is to provide an unprecedentedly detailed understanding of how polymyxin B functions.

We have edited the text as suggested (please see lines: 65-67).

4. The data (Fig. 1D) for mCherry release is less compelling than the other data. Most notably, the well-designed, time-resolved approach provides a key way of linking those processes to which polymyxin lethality can be attributed. Here the data is characterised by a much higher variance than that provided by other methods, and a significant effect is only observed at 90 and 120 mins after the PmB challenge begins. In the same conditions a 2log₁₀ reduction in CFU is already achieved at 30 mins (Fig. 1A). Although the Sytox dye data shows an effect for PmB against stationary phase bacteria in minimal media (which is perhaps not expected from the time-kill data) the timing of the effect of PmB when glucose is added, i.e. at 30 mins connects the Sytox effect and the outcome much better. I wonder if the mCherry release coming at 90 mins is just reflecting death of the cells and provides less insight into mechanism than is asserted?

We agree that such an alternative interpretation (cell death/lysis playing a role in the observed mCherry release) cannot be excluded and have amended the text accordingly (please see lines 102-105).

5. NPN uptake is a very indirect measure of lipid asymmetry or disorder. Can NPN uptake measurements really be used as a proxy for such parameters or should another term be used?

We have revised the text to make clear that NPN is indicative of OM disruption, rather than asymmetry (please see line: 137).

6. The effect of the G907 inhibitor is interesting. Is it possible that it is LPS trafficking rather than synthesis that is hijacked by PmB (trafficking of course follows from synthesis, so this is moot)? Do you see protrusions with this inhibitor or has this not been tested yet?

In response to the reviewers question, we have carried out additional experiments on the G907 inhibitor (now included as part of Supplementary Fig. S12 and mentioned on lines 205-207). As seen for experiments with the LpxC inhibitor CHIR-090, we see a delayed and reduced appearance of protrusions.

7. In Supplementary Figure 5B both *A. baumannii* AS and *E. coli* CFT073 appear to have CFU reductions with glucose but are listed as not significant i.e. $p > 0.05$ with ANOVA with one of four possible tests (as described in the statistical analysis). I wonder if a Brown-Forsythe or Welch test would be appropriate here since the CFU data may have unequal variance (even though log transformed) because of different dilutions needed to record higher or lower CFU values?

We thank the reviewer for this suggestion. We have revised the presentation and analysis of these data and find that a two-way ANOVA confirms significant differences for CFT073 and *A. baumannii* AS.

Reviewer #3 (Remarks to the Author):

It was a pleasure to review the present manuscript titled "Polymyxin B lethality requires energy-dependent outer membrane disruption." This study reports that the lethality of PmB is energy-dependent, and that this is associated with the active transport of LPS to the OM, ultimately leading to blebbing and disruption, allowing exposure of the IM to permeabilizing PmB. The experiments are well controlled, clearly described, and the manuscript is very well written. Although the exact mechanism of LPS shedding is not elucidated in this study, I believe that the observations they have reported reveal a clinically and biologically important paradigm shifting phenomenon that, similar to other classes of antimicrobial drugs, antimicrobial peptides like PmB are only effective against actively growing cells at clinically relevant concentrations. The observations described in this study now make it possible to explore the energy-dependent mechanisms of PmB lethality, which were not previously known to exist. I believe this study is of broad interest to clinical researchers, microbiologists, cell membrane biologists and even bioengineers, and I recommend its publication with very minor revisions. Below I suggest that the authors provide some discussion of the experimental limitations of the study and speculate more deeply on the possible mechanisms of LPS release. To be clear, I do not think any more experiments need to be performed for this study to be published.

The manuscript has been revised to include some speculation on the mechanism(s) underpinning LPS release (please see lines: 405-415), as well as limitations of the work (please see lines: 448-455; 476-479).

Points for additional discussion:

1. It would be interesting to speculate more broadly about the actual mechanism of LPS release (e.g. blebbing, vesiculation, spontaneous shedding) and some questions that could be asked to address this.

The discussion has been revised to include discussion of this point (please see lines 405-415).

2. The role of stress pathways, in particular envelope stress response (e.g. RpoE etc..) might play in responding to or influencing PmB-induced OM remodeling.

We have included discussion of how envelope stress systems may contribute to bacterial survival during PmB exposure and highlighted that there is much more to learn in this area (please see lines: 448-455).

3. What effect could PmB remodeling have on the bacterial immune evasion through its effect on OM remodeling?

We have revised the manuscript to include discussion on the impact of PmB on interactions with host immune components (please see lines: 479-481).